# Decentralized Blockchain-based Robust Multi-agent Multi-armed Bandit

## Abstract

We study a robust, i.e. in presence of malicious participants, multi-agent multi-armed bandit problem where multiple participants are distributed on a fully decentralized blockchain, with the possibility of some being malicious. The rewards of arms are homogeneous among the honest participants, following time-invariant stochastic distributions, which are revealed to the participants only when certain conditions are met to ensure that the coordination mechanism is secure enough. The coordination mechanism's objective is to efficiently ensure the cumulative rewards gained by the honest participants are maximized. To this end and to the best of our knowledge, we are the first to incorporate advanced techniques from blockchains, as well as novel mechanisms, into such a cooperative decision making framework to design optimal strategies for honest participants. This framework allows various malicious behaviors and the maintenance of security and participant privacy. More specifically, we select a pool of validators who communicate to all participants, design a new consensus mechanism based on digital signatures for these validators, invent a UCB-based strategy that requires less information from participants through secure multi-party computation, and design the chain-participant interaction and an incentive mechanism to encourage participants' participation. Notably, we are the first to prove the theoretical regret of the proposed algorithm and claim its optimality. Unlike existing work that integrates blockchains with learning problems such as federated learning which mainly focuses on optimality via computational experiments, we demonstrate that the regret of honest participants is upper bounded by $\log T$ under certain assumptions. The regret bound is consistent with the multi-agent multi-armed bandit problem without malicious participants and the robust multi-agent multi-armed bandit problem with purely Byzantine attacks which do not affect the entire system.

## 1 Introduction

Multi-armed Bandit (MAB) (Auer et al., 2002a;b) models the classical sequential decision making process where a player selects one arm from multiple arms and observes the reward of the pulled arm at each time step. The player aims to maximize the cumulative reward throughout the game, equivalent to the so-called regret minimization problem navigating the trade-off between exploration (e.g., exploring unknown arms) and exploitation (e.g., favoring the currently known optimal arm). The recent emerging advancement of federated learning, wherein multiple participants jointly train a shared model, has spurred a surge of interest in the domain of multi-agent multi-armed bandit (multi-agent MAB). In this context, multiple participants concurrently interact with multiple MABs, with the objective being the optimization of the cumulative averaged reward across all the participants through communications. Significantly, these participants face additional communication challenges.

Numerous research has been focused on the multi-agent MAB problem, including both centralized settings as in (Bistritz and Leshem, 2018; Zhu et al., 3–4, 2021; Huang et al., 2021; Mitra et al., 2021; Réda et al., 2022; Yan et al., 2022), and decentralized settings as in (Landgren et al., 2016a;b; 2021; Zhu et al., 2020; Martínez-Rubio et al., 2019; Agarwal et al., 2022), where it is assumed that reward distributions are uniform among participants, namely homogeneous. Recent attention has shifted towards addressing decentralized, heterogeneous variants, including (Tao et al., 1546–1574, 2022; Wang et al., 1531–1539, 2021; Jiang and Cheng, 1–33, 2023; Zhu et al., 2020; 2021; 3–4, 2021; Zhu and Liu, 2023; Xu and Klabjan, 2023b), which are more general and bring additional complexities. In these scenarios, the shared assumption is that all participants exhibit honesty, refraining from any

malicious behaviors, and adhere to both the shared objective and the designed strategies. However, real-world scenarios often deviate from this assumption, are composed of malicious participants that perform disruptively. Examples include failed machines in parallel computing, the existence of hackers in an email system, and selfish retailers in a supply chain network. Consequently, recent research, such as (Vial et al., 2021; Zhu et al., 2023), has focused on the multi-agent MAB setting with malicious participants, which is formulated as a robust multi-agent MAB problem. This line of work yields algorithms that perform optimally, provided that the number of malicious participants remains reasonably limited.

However, there are three major concerns related to the existing robust multi-agent MAB framework, namely optimality, security, and privacy, respectively. Firstly, in (Vial et al., 2021), the truthfulness of the integrated reward estimators by participants is not taken into account. Every participant maintains reward estimations and thus we also call them estimators. In essence, it means it might not be possible to assert the correctness of these estimators, even though the relative differences between the arms are bounded. In certain scenarios, estimators play a crucial role in guiding decision making. For instance, in the context of smart home (Zhao et al., 2020), driven by the rapid growth of the Internet of Things (IoT), in a smart home device setting the suppliers of the devices are the participants monitored by the manufacturer, the devices are the arms, and the consumers are the environment, the manufacturer seeks to understand consumer behavior. The reward corresponds to any metric measuring consumer engagement. Each supplier develops its own engagement (reward) estimatation by arm pulls where it is important for the estimators to be accurate. The knowledge about the ground truth, i.e. consumer behavior in expectation across time, is essential, making the correctness of estimators a critical concern. Secondly, there is the possibility of malicious participants (suppliers) exhibiting various disruptive behavior beyond broadcasting inaccurate estimators which is a facet not covered in existing work (Vial et al., 2021; Zhu et al., 2023). For example, in a network routing problem, where devices (i.e., participants) send information through communication channels that represent the arms to maximize information throughput (i.e., the reward), malicious participants could intentionally cause channel congestion and disrupt the traffic that honest participants rely on. This has the potential to systematically affect the performance of honest participants as a significant motivation herein. Thirdly, existing literature assumes that participants are willing to share all the information with other participants, including the number of pulls of arms and the corresponding reward estimators. This, however, exposes the participants to the risk of being less private, as it might be easy to retrieve the cumulative reward and action sequence, based on the shared information. This has not yet been explored by the existing work and thereby motivating our work herein.

Notably, blockchains have a great potential to address these challenges, which are fully decentralized structures and have demonstrated exceptional performance in enhancing system security and accuracy across a wide range of domains (Feng et al., 2023). This trending concept, widely applied in finance, healthcare and edge computing, was initially introduced to facilitate peer-to-peer (P2P) networking and cryptography, as outlined in the seminal work by (Nakamoto, 2008). A blockchain (permissioned where the set of participants is fixed versus permisionless where the set of participants is dynamic and anyone can join) comprises of a storage system for recording transactions and data, a consensus mechanism for participants to ensure secure decentralized communication, updates, and agreement, and a verification stage to assess the effectiveness of updates, often referred to as block operations (Niranjanamurthy et al., 2019), which thus provides possibilities for addressing the aforementioned concerns. First, the existence of verification guarantees the correctness of the information before adding the block to the maintained chain, and the storage system ensures the history is immutable. Secondly, the consensus mechanism ensures that honest validators, which are representatives of participants, need to reach a consensus even before they are aware of each other's identities, leading to a higher level of security and mitigating systematic attacks. Lastly, enabling cryptography and full decentralization without a central authority has the potential to improve the privacy level. However, no work has studied how blockchains can be incorporated into an online sequential decision making regime, creating a gap between multi-agent MAB and blockchains that we take a step to close it.

There has been a line of work adapting blockchains into learning paradigms, and blockchain-based federated learning has been particularly successful as in (Li et al., 2022; Zhao et al., 2020; Lu et al., 2019; Wang et al., 2022). In this context, multiple participants are distributed on a blockchain, and honest participants aim to optimize the model weights of a target model despite the presence of malicious participants. Notably, the scale of the model has led to the introduction of a new storage system on the blockchain, the Interplanetary File System (IPFS), which operates off-chain, ensuring

the stability and efficiency of block operations on the chain. However, due to the unique decision making in MAB, the existing literature does not apply to the multi-agent MAB, necessitating a novel framework for blockchain-based multi-agent MAB. Moreover, there is limited study on the theoretical effectiveness of blockchain-based federated learning, as most studies focus on their deployment performances. Theoretical validity is crucial to ensure cybersecurity because deploying blockchains, even in an experimental setting, is risky and has been extremely challenging. Henceforth, it remains unexplored how to effectively incorporate blockchains into the robust multi-agent MAB framework, and how to analyze the new algorithms theoretically, which we address herein.

To this end, we herein propose a novel formulation of robust multi-agent MAB on blockchains. Specifically, we are the first to study the robust multi-agent MAB problem where participants are fully distributed, can be malicious, and operate on permissioned blockchains. In this context, a fixed set of participants pull arms and communicate to validators, and validators communicate with one another and decide on a block to be sent to the chain. Participants can only receive rewards when the block is approved, in order to ensure security, which means the rewards are conditionally observable, even for the pulled arm, which complicates the traditional bandit feedback and introduces new challenges. Participants can be malicious in various disruptive aspects. The objective of the honest participants is to maximize their averaged cumulative received reward. Participants not only design strategies for selecting arms but also strategically interact with both other participants and the blockchain. The blockchain keeps track of everything (the history is immutable), guarantees the functionality of the coordination mechanism through chain operations, and communicates with the environment.

For the new formulation, we develop an algorithmic framework, motivated by existing work while introducing novel techniques. The framework uses a burn-in and learning period. We incorporate a UCB-like strategy into the learning phase to perform arm selection, while using random arm selection during the burn-in period. We also use validator/commander selection to eliminate the need for an authorized leader, including both full decentralization and efficient reputation-based selection. We propose the update rules for both participants and validators to leverage the feedback from both the environment and the participant set. Furthermore, we modify the consensus protocol without relying on $\frac{2}{3}$ voting; instead, we use a digital signature scheme (Goldwasser et al., 1988) coupled with the consensus protocol in (Lamport et al., 2019). Moreover, we introduce the role of a smart contract (Hu et al., 2020) that interacts with both the blockchain and the environment, which validates the consensus and collects the feedback from the environment. To incentivize malicious participants (we want the malicious participants to actively participate via information sharing in order to be identified soon) we invent a novel cost mechanism inspired by the use of mechanism design in federated learning (Murhekar et al., 2023). It is worth noting that the existence of this smart contract and cost mechanism also guarantees the correctness of the information transmitted on the chain.

Subsequently, we perform theoretical analyses of the proposed algorithm. We formally analyze the regret that reflects optimality and fundamental impact of malicious behavior on blockchains. Precisely, we show that under different assumptions in different settings, the regret of honest participants is always upper bounded by $O(\log T)$, consistent with existing robust multi-agent MAB (Zhu et al., 2023; Vial et al., 2021). This is the very first theoretical result on leveraging blockchains for online sequential decision making problems, to the best of our knowledge. Furthermore, this regret bound coincides with the existing regret lower bounds in multi-agent MAB when assuming no participants are malicious (Xu and Klabjan, 2023a), implying its optimality. We also find that, surprisingly, various aspects about security are by-products of optimality.

Our main contributions are as follows. First, we propose a novel formulation of multi-agent MAB with malicious participants, where rewards are obtainable only when the coordination mechanism's security is guaranteed. Additionally, the actual received rewards account for the accuracy of the shared information through our proposed cost mechanism. To maximize the cumulative rewards of honest participants, we develop a new algorithmic framework that introduces blockchain techniques. Along the way we design new mechanisms and protocols. We also prove the theoretical effectiveness of the algorithm through an extensive analysis of regret under assumptions on the problem setting, such as the ratio of honest participants, the cost definition, and the validator selection protocol. This work bridges the gap between cybersecurity and online sequential decision making.

The structure of the paper is as follows. In Section 2, we introduce the problem formulation and notations. In Section 3, we propose the algorithmic framework. Subsequently, in Section 4, we provide detailed analyses of the theoretical guarantee for the proposed algorithms.

## 2 PROBLEM FORMULATION

We start by introducing the notations used throughout the paper. Consistent with the traditional MAB setting, we consider $K$ arms, labeled as $1, 2, \ldots, K$. The time horizon of the game is denoted as $T$, and let us denote each time step as $1 \leq t \leq T$. Additionally as in Multi-agent MAB, let us denote the number of participants as $M$ labeled from 1 to $M$. We denote the public and secret keys of participant $m$ as $(PK_m, SK_m)$ for any $1 \leq m \leq M$. The list of public keys $PK_1, PK_2, \ldots, PK_M$ is public to anyone, in the order indicated by the participant set. Meanwhile, in our newly proposed blockchain framework, we denote the total number of blocks as $B = T$ and whether each block at time step $t$ is approved or not is represented by a binary variable $b_t \in \{0, 1\}$. Let us denote the reward of arm $i$ at participant $m$ at time step $t$ as $\{r_i^m(t)\}_{i,m,t}$, which follows a stochastic distribution with a time-invariant mean value $\{\mu_i\}_i$. Let $a_m^t$ be the arm selected at time $t$ by participant $m$ and let $n_{m,i}(t)$ be the number of arm pulls for arm $i$ at participant $m$ at time $t$. We denote the set of honest participants and malicious participants as $M_H$ and $M_A$, respectively, which are not known apriori. Note that they are time-invariant. Similarly, let $S_V(t)$ denote the set of validators at time $t$ which is algorithmically determined. We denote the estimators maintained at participant $m$ as $\bar{\mu}_i^m(t), \tilde{\mu}_i^m(t)$ for local and global reward estimators, respectively, and the validators estimators as $\tilde{\mu}_i(t)$. We point out that $\tilde{\mu}_i(t)$ is a function of $\bar{\mu}_i^m(t)$. What are stored in a block is deferred to Appendix F.

The process during one iteration is as follows. At the beginning of each decision time, each participant selects an arm based on its own policy. Then, a set of validators is selected, and the participants broadcast their reward estimators to the validators. The validators perform aggregation of the collected information. Next, they run a consensus protocol to examine whether the majority agree on the aggregated information, a process called validation. They send the validated information to the smart contract, which verifies its correctness and sends feedback to the environment. If the smart contract is approved, the blockchain is updated. Lastly, the environment distributes the reward information plus cost based on the feedback from the smart contract (only if the block has been approved). The participants then update their estimators accordingly. The corresponding flowchart is presented in Figure 1.

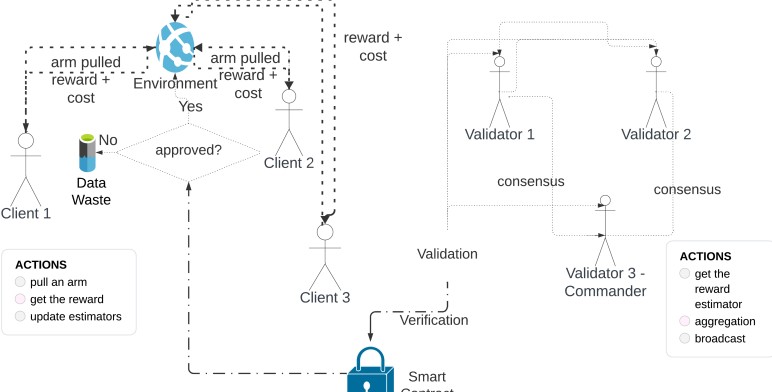

Figure 1: The flow of the algorithm

**Cost Mechanism** We propose a cost mechanism where if the estimators from the malicious participants are used in the validated estimators, i.e. $\frac{\partial \tilde{\mu}_i(t)}{\partial \bar{\mu}_i^m(t)} \neq 0$, then the honest participants incur a cost of $c_t^m \geq 0$ and malicious participants receive $c_t^m < 0$, which they are not aware of until the end of the game. It incentives the participation of malicious participants, in particular, given that they may not be willing to share anything. In the meantime, as a by-product, it also penalizes the aggregated estimators by the honest participants, which ensures the correctness of the estimators. In addition $c_t^m = 0$ for every $m$ if $\frac{\partial \tilde{\mu}_i(t)}{\partial \bar{\mu}_i^m(t)} = 0$.

With the goal to maximize the total cumulative (expected) reward of honest participants, we define the regret as follows. We denote the cumulative reward of honest participants as $r_T = \sum_{m \in M_H} \sum_{t=1}^{T} r_{a_m^t}^m(t) 1_{b_t=1} - \sum_{t=1}^{T} c_t$ and the regret as $R_T = \max_i \sum_{m \in M_H} \sum_{t=1}^{T} r_i^m(t) 1_{b_t=1} - r_T$ and pseudo regret $\bar{R}_T = \max_i \sum_{m \in M_H} \sum_{t=1}^{T} \mu_i^m - E[r_T]$ (rationality in Appendix F).

Attacking principles are described in Appendix D (Existential Forgery, Adaptive Chosen Message Attack, and Universal Composability Framework), which are used in the digital signature scheme (Goldwasser et al., 1988) and secure multi-party computation (Asharov et al., 2012),

## 3 METHODOLOGIES

In this section, we present our proposed methodologies within this new framework. Notably, we develop the first algorithmic framework at the interface of blockchains in cybersecurity and multi-agent MAB in online sequential decision making, addressing the joint challenges of optimality, security, and privacy. We leverage the blockchain structure while introducing new advancements to the existing ones, to theoretically and efficiently guarantee the functionality of the chain with new consensus protocols and a cost mechanism. Additionally, it is designed for online sequential decision making scenarios, distinguishing our work from existing literature on federated learning. Moreover, compared to existing work on Byzantine-resilient multi-agent MAB, our methodology operates on a blockchain with an added layer of security and privacy.

In the framework, every participant can be either honest or malicious, including selected validators and commanders. The number of malicious participants must be known apriori, but not who are they. The malicious clients can perform every step in an adversary manner to their likening except updating trust coefficients $(p_m, w_m)$. For this step, they must follow the general agreed-upon rule. This assumption is quite common in blockchain works. The only component that must know who is malicious and who is honest is the environment (since it must assign appropriate penalties based on these designations). In our use case of smart homes, this means that the customers know which supplier is malicious and which supplier is honest (for example based on google reviews). The algorithmic framework is composed of two phases: the burn-in period, which is a warm-up phase for $t \leq L$, where $L$ is the length of the burn-in period, and the learning period, where $t > L$. It consists of 5 functions, with the main algorithm presented in Algorithm 1, and the remaining functions detailed in Algorithms 2-5 (Appendix C). Algorithm 1 constitutes the core of the methodology, including the sequential strategies executed by the honest participants, black-box operations by the malicious participants, and the chain executions.

The core algorithm includes several stages, as indicated in the following order. We present the pseudo code of the core algorithm in Algorithm 1, named BC-UCB. Here, the common random seed $\bar{q}_t$, $t = 1, \ldots, T$ and the random seed for each participant $q = (q_1, q_2, \ldots, q_M)$ are publicly known in advance. Function $VRF$ refers to verifiable random functions proposed in (Micali et al., 1999) composed of $G$ that represents the generating function for the public and secret key with seed $q^0$, i.e. $G(q^0) = (pk, sk)$, and $VRF_F(\bar{q}_t, G(q^0)) = (hash, \pi)$ where $hash$ is a hash value and $\pi$ is a function (proof) that returns $True$ or $False$ given $hash$ and public key $pk$, i.e. $\pi(hash, pk)$ outputs $True$ or $False$. Let $hl$ be the size of $hash$ which is an input to $\pi$. For any multi-set $S_0$, $majority(S_0)$ refers to an element in $S_0$ with the highest count.

**Arm selection** As in an MAB framework, the participants decide which arm to pull at each time step. The strategies depend on whether participants are honest or malicious. The honest participants follow a UCB-like approach. More specifically, each honest participant $m$ selects arm $a_m^t = t \mod K$ during the burn-in period. During the learning period, it assigns a score to each arm $i$ and selects the arm with the highest score, which can be formally written as $a_m^t = \arg\max_i \tilde{\mu}_i^m(t-1) + F(m, i, t-1)$ where $\tilde{\mu}_i^m(t)$ is the maintained estimator at participant $m$. Here $F(m, i, t) = (\frac{C_1 \log t}{n_{m,i}(t)})^\beta$ with constant $C_1, \beta$ being specified in the theorems. A malicious participant $j$, however, selects arms based on arbitrary strategies, which is also known as Byzantine's attack and written as $a_j^t = h_j^t(\mathcal{F}_t) \in [K]$ where $\mathcal{F}_t$ denotes the history up to time step $t$ (everything on the blockchain and additional information shared by other participants).

**Validator or Commander Selection** At each time step, a coordination mechanism or iterative protocol selects a pool of participants allowed to act on the chain, known as validators. The commanders are selected in the similar way but with different parameters. Specifically, the coordination mechanism samples the set of validators and commanders according to Algorithm 2 (in Appendix C), based on the trust coefficients of participants $p_m(t), w_m(t)$. The chain relays this set of validators to aggregate the reward estimators and to achieve consensus as detailed below. The commanders participate in the consensus protocol.

We use a smart contract that takes membership of a participant in $S_V(t)$ as input and produces a single sorted list of $S_V(t)$ based on the public keys $PK_m$ of participants. It is worth noting that the sorting function can be incorporated into the script. Then, the participants access this smart contract $sc_{sort}$ with input $S_V(t), PK$ to obtain the sorted list of validators $S_V(t)$ from its output.

**Broadcasting** During broadcasting, the participants sent information to validators which then perform the aggregation step. To expand, malicious participant $j$ broadcasts its estimators $\bar{\mu}_i^j(t)$ to the validators using a black-box attack, e.g. a Byzantine's attack or a backdoor attack. Honest participant $m$ broadcasts its true reward estimators $\bar{\mu}_i^m(t)$ to the validators.

**Aggregation** Next, the validators integrate the received information. Specifically, for each honest validator $j$, an honest validator determines the set, $A_t^j, B_t^j$ as follows. For $t > L$, the set $A_t^j$ reads as $m \in A_t^j \Leftrightarrow n_{m,i}(t) > \frac{n_{j,i}(t)}{k_i(t)}$ for every $i$ where $k_i(t) \geq \max_{k \in M} \frac{n_{k,i}(t)K}{L}$ is the threshold parameter which can be constructed through the secure multi-party computation protocol as in (Asharov et al., 2012), without knowing the value of $n_{m,i}(t)$ to ensure privacy. More specifically, each participant $m$ sends $n_{m,i}(t)$ and the value of $k_i(t)$ to the protocol. The protocol then outputs whether $m \in A_t^j$. The set $B_t^j$ is computed as follows, depending on the size of $A_t^j$. If $|A_t^j| > 2f$ where $f = |M_A|$ and the process is in the learning period $t > L$, then $B_t^j = \cup_i\{(m, \bar{\mu}_i^m(t)) : \bar{\mu}_i^m(t)$ is smaller than the top $f$ values in $A_t^j$ and larger than the bottom $f$ values in $A_t^j\}$. Otherwise in burn-in, $B_t^j = \{t \bmod K\}$ and $A_t^j = \emptyset$. Once again, the malicious participants choose the sets $A_t$ and $B_t$ in a black-box manner.

**Consensus** The consensus protocol is central to the execution of the blockchain and guarantees that the chain is secure. More specifically, we incorporate the digital signature scheme (Goldwasser et al., 1988) into the solution to the Byzantine General Problem (Lamport et al., 2019) under any number of malicious validators. The pseudo code is presented in Algorithm 3. First, a commander is selected from the validators that broadcasts $B_t$ to other validators with its signature generated by (Goldwasser et al., 1988), which we call a message. This process is then repeated at least $M$ times, based on the algorithm in (Lamport et al., 2019). The validators output the mode of the maintained messages. The consensus is successful if more than 50% of the validators output the $B_t$ maintained by the honest validators. Otherwise, the consensus step fails, resulting in an empty set $B_t$.

**Global Update** The set $B_t$ is then sent to the validators, which compute the average of the estimators within $B_t$, known as the global update detailed in Algorithm 4. More precisely, for each arm $i$ at time step $t$, the estimator is computed as $\tilde{\mu}_i(t) = \frac{1}{2}(\hat{\mu}_i(t) + \tilde{\mu}_i(\tau)), \hat{\mu}_i(t) = \frac{\sum_{m \in B_t} \bar{\mu}_i^m(t-1)}{|B_t|}$ where $\tau = \max_{s<t}\{b_s = 1\}$. If $B_t$ is not empty, and $\tilde{\mu}_i(t) = \infty, \hat{\mu}_i(t) = \infty$, otherwise.

**Block Verification** The validators run the smart contract $sc_{block}$ to validate the block and assign $b_t = 1$ if the estimator satisfies the condition $\tilde{\mu}_i(t) \leq 2$. It disapproves the block otherwise, which is denoted as $b_t = 0$.

**Block Operation** At the beginning of the algorithm, the environment sets a random cost value $c_t = c$ between 0 and 1. The smart contract sends the output containing the validated estimator $\tilde{\mu}_i(t)$, the set $B_t$, and the indicator $b_t$ of whether the block is approved to the environment. Subsequently, the environment determines the rewards, namely Block Operation, as in Algorithm 5, to be distributed to the participants based on the received information from the smart contract, in the following three cases. **Case 1**: If $b_t = 1$ and $B_t \subset M_H$, i.e $\frac{\partial \tilde{\mu}_i(t)}{\partial \bar{\mu}_i^m(t-1)} = 0$ for every $m \notin M_H$, then the environment distributes $r_{a_m^t}^m(t)$ and $\tilde{\mu}_i(t)$ to participant $m$ for any $1 \leq m \leq M$. **Case 2**: If $b_t = 1$ and $B_t \cap M_H < |B_t|$, i.e. there exists $m \in M_A$ such that $\frac{\partial \tilde{\mu}_i(t)}{\partial \bar{\mu}_i^m(t-1)} \neq 0$, then the environment distributes $r_{a_m^t}^m(t) - c_t$ and $\tilde{\mu}_i(t)$ to any honest participant $m$, and $r_{a_j^t}^j(t) + c_t$ to any malicious participant $j$. **Case 3**: If $b_t = 0$, the environment distributes nothing to the participants.

**Participants' Update** After receiving the information from the environment, the honest participants update their maintained estimators as follows. **Rule** For the global reward estimator $\tilde{\mu}_i^m(t)$, they update it when they receive $\tilde{\mu}_i(t)$, i.e. $\tilde{\mu}_i^m(t) = \tilde{\mu}_i(t)$, and otherwise, $\tilde{\mu}_i^m(t) = \bar{\mu}_i^m(t)$. For the number of arm pulls and the local reward estimators, they update them as $n_{m,i}(t) = n_{m,i}(t-1) + 1_{b_t=1} \cdot 1_{a_m^t=i}, \bar{\mu}_i^m(t) = \frac{\bar{\mu}_i^m(t-1) + r_{a_m^t}^m(t) \cdot 1_{a_m^t=i}}{n_{m,i}(t)}. \ (1)$

In SELECTION the value $w_m(t)$ implies a certain number of commanders, i.e. the number of commanders is a function of $w_m(t)$ and likewise the validators with respect to $p_m(t)$. We call $p_m, w_m$ trust coefficients. Based on the concept of staking, $p_m(t) = \frac{\sum_{s \leq t} r_{a_s}^m(s)}{\sum_{m=1}^{M} \sum_{s \leq t} r_{a_s}^m(s)}$.

---

**Algorithm 1** BC-UCB

---

Initialization: For participants $1, 2, \ldots, M$, arms $1, \ldots, K$, at time step $0$ we set $\tilde{\mu}_i^m(0) = \hat{\tilde{\mu}}_i(0) = n_{m,i}(1) = 0$; the number of honest participants $M_H$; Verifiable Random Function $VRF$

**for** $t = 1, 2, \ldots, T$ **do**
    **for** *each participant $m$* **do**                             // Validator Selection
        Sample $z = \text{SELECTION}(t, m, p_m(t-1), VRF)$. If $z = 1$, participant $m$ is a validator.
    **end**
    Let $S_V(t)$ be the set of all validators
    **for** *each validator $m$* **do**               // Commander Selection $(w_m = \frac{1}{|S_V(t)|})$
        Sample $z = \text{SELECTION}(t, m, \frac{1}{|S_V(t)|}, VRF)$. If $z = 1$, validator $m$ is a commander.
    **end**
    Let $S_C(t)$ be the set of commanders
    **for** *each participant $m \in M_H$* **do**               // Arm Selection – UCB
        **if** *$k \in A_t^m$ for every $k \in M_H$ with $S_V(t)$ and $S_C(t)$ and $t > L$* **then**
            $a_m^t = \arg\max_i \tilde{\mu}_i^m(t-1) + F(m, i, t-1)$
        **else**
            Sample an arm $a_m^t = t \mod K$.
        **end**
        Pull arm $a_m^t$

    **end**
    **for** *each participant $m \in M_A$* **do**          // Arm Selection – Any Strategy
        Select an arm $a_m^t$ and pull arm $a_m^t$

    **end**
    **for** *each participant $m$* **do**                       // Broadcasting
        Broadcast $\bar{\mu}_i^m(t-1)$ to validators $S_V(t)$, where malicious participants $m \in M_A$ use an
        attack regarding an arm $a_m^t$, i.e., $\bar{\mu}_i^m(t-1) = \bar{h}_{m,i}^t(\mathcal{F}_{t-1})$.
    **end**
    **for** *each participant $m \in S_V(t)$* **do**                // Aggregation
        Validator $m \in M_H \cap S_V(t)$ determines the set $B_t^m = B_t$ containing trusted participants $j$
        and the corresponding estimators $\bar{\mu}_i^j(t)$
        Validator $m \in M_A, m \in S_V(t)$ arbitrarily determines the set $B_t^m$
    **end**
    // Consensus
    Validators run consensus on $\{B_t^m\}_m$ according to $\text{CONSENSUS}(S_C(t), \{B_t^m\}_m, M)$
    Validators run the smart contract $sc_{block}$ to compute $\tilde{\mu}_i(t)$ according to $\text{GLOBAL\_UPDATE}(B_t)$
    // Global Update
    Validators perform Block Validation:               // Block Verification
      **If** there exists $i \in \{1, 2, \ldots, K\}$ with global estimator $\tilde{\mu}_i(t) < \infty$
        Approve the block by letting $b_t = 1$
      **else**
        Disapprove the block by letting $b_t = 0$
      **end**
    // Environment
    The environment sends rewards to participants using $\text{OPERATION}(\tilde{\mu}_i(t), a_{m,m}^t, B_t, b_t)$
    **for** *each participant $m$* **do**                   // participants' Update
        Participant $m \in M_H$ updates $\tilde{\mu}_m(t), n_{m,i}(t), \bar{\mu}_m(t), p_m(t), w_m(t)$ based on Rule; partici-
        pant $m \in M_A$ updates $\tilde{\mu}_m(t), n_{m,i}(t), \bar{\mu}_i^m(t)$ arbitrarily
    **end**
**end**

---

# 4 REGRET ANALYSES

In this section, we demonstrate the theoretical guarantee of our proposed framework by conducting a comprehensive study of the regret as in Section 2. Specifically, to allow for flexibility and generalization, we consider various problem settings, including the number of malicious participants $M_A$, the cost definition $c_t$, the commander selection rule, and the validator selection rule.

The initial set of results does not consider $p_m$ and $w_m$; instead, they impose assumptions on the number of validators and honest participants. They should be interpreted as letting $p_m$ and $w_m$ be such that the outcome of Validator/Commander Selection (Algorithm 2 in Appendix C) has the desired properties. All proof steps are in Appendix E.

### 4.1 Limited Numbers of Malicious Participants

Most existing work on blockchain or robust optimization (Nojoumian et al., 2019; Feng et al., 2023; Li et al., 2022) considers a limited number of malicious participants, as majority voting-based consensus and the accuracy of the constructed estimators largely depend on whether the ratio of malicious participants is reasonable. An extreme case occurs when all but one participant are malicious, rendering the methods in this literature inapplicable, since the blockchain cannot achieve consensus. Therefore, consistently, we first analyze the regret bound given a limited number of participants.

#### 4.1.1 Low numbers of malicious participants and constant cost

First, we consider the case when the number of honest participants is larger than $\frac{2}{3} \cdot M$ which is the same as $M_A \le \frac{1}{3}$, and there are $\frac{1}{3}M + 1$ commanders. The cost mechanism uses constant cost, i.e. $c_(t) = c$ where $c$ is specified in Section 3, which requires honest participants to exclude any estimator that is from the malicious participants when updating $\hat{\mu}_i(t)$. Meanwhile, commander selection assumes that at least one honest participant serves as a commander, which allows the honest participants to achieve consensus on the accurate $\tilde{\mu}$. The formal statement reads as follows.

**Theorem 1.** *Let us assume that the total number of honest participants is at least $\frac{2}{3}M$ and that there is at least one honest participant in the validator set. Meanwhile, let us assume that the malicious participants perform existential forgery on the signatures of honest participants with an adaptive chosen message attack. Lastly, let us assume that the participants are in a standard universal composability framework when constructing A. Then we have that $E[R_T|A] \le (c+1) \cdot L + \sum_{m \in M_H} \sum_{k=1}^K \Delta_k([\frac{4C_1 \log T}{\Delta_i^2}] + \frac{\pi^2}{3}) + |M_H|Kl^{1-T}$ where $L$ is the length of the burn-in period of order $\log T$, $c > 0$ is the cost, $C_1$ meets the condition that $\frac{C_1}{6|M_H|k_i\sigma^2} \ge 1$, $\sigma^2 \ge \frac{1}{M_H}$, $\Delta_i$ is the sub-optimality gap, $l$ is the length of the signature of the participants, and $k_i$ is the threshold parameter used in the construction of $A_t$. Here the set $A$ is defined as $A = \{\forall 1 \le t \le T, b_t = 1\}$ which satisfies that $P(A) \ge 1 - \frac{1}{l^{T-1}}$.*

*Proof sketch.* The full proof is provided in Appendix E; the main logic is as follows. We decompose the regret into three parts: 1) the length of the burn-in period, 2) the gap between the rewards of the optimal arm and the received rewards, and 3) the cost induced by selecting the estimators of the malicious participants. For the second part of the regret, we bound it in two aspects. First, we analyze the total number of times rewards are received, i.e., when the block is approved, which is of order $1 - l^{1-T}$. Then, we control the total number of times sub-optimal arms are pulled using our developed concentration inequality for the validated estimators sent for verification. Concerning the third part, we bound it by analyzing the construction of $B_t$, which depends on the presence of malicious participants in $A_t$. By demonstrating that $A_t$ contains only a small number of malicious participants in comparison to the total number of honest participants, we show that $B_t$ does not induce additional cost. Combining the analysis of these three parts, we derive the regret bound. □

#### 4.1.2 Moderate number of malicious participants and distance-based cost

Along the line of work on robust optimization (Dong et al., 2023), a common assumption is that at least $\frac{1}{2}$ participants are honest. To this end, we relax the assumption on the minimal number of honest participants from $\frac{2}{3}$ to $\frac{1}{2}$, modifying the cost definition. We propose the following algorithmic changes, an alternative to the aggregation step. We call the already proposed strategy as Option 1.

**Option** 2 Construct a filter list $A_t$ as in Option 1. Construct a block list $B_t \subset A_t$ for any $t > L$ as $B_t = \{m : \bar{\mu}_i^m(t)$ is smaller than the top $f$ values and larger than the bottom $f$ values$\}$.

The choice of the option affects step 2 in Aggregation. Let us assume that sets $A_t, B_t$ are constructed based on Option 2, instead of Option 1 in Theorem 1. We also need to adjust the global estimator $\tilde{\mu}_i(t)$ in Global Update as $\tilde{\mu}_i(t) = P_t\tilde{\mu}_i(t-1) + (1 - P_t)\hat{\mu}_i(\tau)$ where $P_t = 1 - \frac{1}{t}$ and again $\tau = \max_{s<t}\{b_s = 1\}$. Finally, the cost associated with the global estimator is constructed as $c_t = \min_i Dist(\tilde{\mu}_i(t), \mu_i)$, where $Dist(\tilde{\mu}_i(t), \mu_i) = |\tilde{\mu}_i(t) - \mu_i|^6$. The length of the burn-in period is now $(\frac{\log T}{2})^{\frac{1}{6}}$. We point out that Operation is executed by the environment which is the only entity having the knowledge of $\{\mu_i\}_i$. The formal regret statement reads as follows.

**Theorem 2.** *Let Option 2 be used. Let us assume that the total number of honest participants is at least $\frac{1}{2}M$ and let us assume that there is at least one honest participant in the validator set. Meanwhile, let us also assume that the malicious participants perform existential forgery on the signatures of honest participants with an adaptively chosen message attack. Lastly, let us assume that the participants are in a standard universal composability framework when constructing $A$. Then we have that $E[R_T|A] \leq (c+1) \cdot L + O(\log T) + |M_H|Kl^{1-T}$ where $L$ is the length of the burn-in period of order $(\log T)^{\frac{1}{6}}$, $c$ is an uniform upper bound on the cost $c_t$, and $l$ is the length of the signature of the participants. Here the set $A$ is defined as in Theorem 1.*

### 4.1.3 LARGE NUMBER OF MALICIOUS PARTICIPANTS AND DISTANCE-BASED COST

Surprisingly, we report next that by more precisely characterizing the different types of malicious behaviors, we can relax the assumption on the number of malicious participants. **Structure of malicious behaviors** We define set $M_A^1 \subset M_A$ as comprising of malicious participants that only perform attacks on the estimators. Furthermore, we denote by $M_A^2 \subset M_A$ the set comprising of malicious participants that perform attacks on the consensus. Also, $M_A^{2,1} \subset M_A^2$ are the malicious participants that perform attacks on both the estimators and the consensus. Note that all the malicious participants are allowed to perform existential forgery on the signatures of the honest participants.

We next introduce Option 3 as an alternative to options 1 and 2. **Option 3** Construct a filter list $A_t$ and the initial $B_t$ as in Option 2. In this option, we further refine $B_t$. If honest participant $m$ is a validator at time step $t$, then it maintains a participant blocklist $D_t$ such that $\{d \in D_t : d \in S_C(t),$ participant $d$ attacks the consensus in that $d$ signs two different messages (the received one from other participants and a self-modified one) and sends the self-modified one$\}$. Let $B_t = B_t \cap (D_t)^c$ where $(D_t)^c$ represents the compliment set of $D_t$. Note that the construction of set $D_t$ is feasible, as the honest participant can track the public key (the signature) through tracing back a Chandelier tree, and thus track the label through the fixed mapping between the participants' public keys and the labels.

**Theorem 3.** *The algorithm is applied with Option 3 and the aforementioned distance-based cost. Let us assume that the total number of honest participants is at least $\frac{1}{4}M$ and let us assume that $M_A^1 < M_H - 1$ and $M_A^2 < \frac{1}{2}M - 1$. Meanwhile, let us assume that the malicious participants perform existential forgery on the signatures of honest participants with an adaptive chosen message attack. Lastly, let us assume that the participants are in a standard universal composability framework when constructing $A$. Then we have that $E[R_T|A] \leq O(\log T)$.*

## 4.2 GENERAL NUMBER OF MALICIOUS PARTICIPANTS

What we have assumed thus far is that there is a limited number of malicious participants. Ideally, one would expect the protocol to accommodate any number of malicious participants. Although (Zhu et al., 2023) addresses this general situation, their work is not related to the blockchain protocol. To this end, we explore this general setting where there can be any number of malicious participants on a blockchain. However, intuitively, if a majority of participants engage in an attack on the consensus, the blockchain can always be invalidated, resulting in a linear regret lower bound. To account for a general number of participants, we refine the structure of malicious behaviors and allow for multiple types of malicious behaviors as an additional assumption in this broader context.

### 4.2.1 GENERAL NUMBER OF MALICIOUS PARTICIPANTS AND DISTANCE-BASED COST

Besides the $\frac{3}{4}$ assumption, more surprisingly, we find that this brand new algorithmic framework works for more general settings with any number of participants, assuming a more refined structure of the malicious participants. The cost definition is again the distance-based one with Option 3.

**Theorem 4.** *Let us assume that the total number of honest participants is arbitrary. Let us assume that $M_A^1 < \frac{1}{2}M - 1$ and $M_A^2 < \frac{1}{2}M - 1$, and further assume that $M_A^{2,1} = \emptyset$. The cost is the distance-based cost. Meanwhile, let us assume that the malicious participants perform existential forgery on the signatures of honest participants with an adaptive chosen message attack. Lastly, let us assume that the participants are in a standard universal composability framework when constructing $A$. Then we have that $E[R_T|A] \leq O(\log T)$.*

### 4.2.2 GENERAL NUMBER OF MALICIOUS PARTICIPANTS WITH AN EFFICIENT COMMANDER SELECTION PROTOCOL

So far, what we have discussed imposes assumptions on the outcome of selection. While this guarantees the decentralization of the coordination mechanism, there is room for improvement in efficiency. As an extension, we consider a more general commander selection procedure in the protocol, with adaptive numbers of commanders, to improve efficiency while ensuring decentralization.

**Commander selection**    The commander set $C_t^s$ is determined by executing Algorithm 2 in Appendix C where the trust coefficients $w_m(t)$ are the probabilities of being selected as commanders. Let $w_m(t) = w_m = 1 - \frac{\log T}{T}$, for any $m \in M_H$ and $w_j(t) = w = \frac{\log \frac{|M_A|}{\eta}}{L}$, for any $j \in M_A$. Subsequently, we establish the following regret bound based on $w_j$ from these two choices and Option 3. Due to the choice of $w_m$, we no longer require that there is at least one honest commander.

**Theorem 5.** *Assume the same conditions as in Theorem 4 where the cost is the distance-based one. Let us assume that the commanders are selected based on the above protocol, and the estimators are computed as aforementioned. We still require that at least one half of the validators are honest. Then we obtain that the regret upper bound with respect to our algorithm is $O(\log T)$.*

### 4.2.3 General number of malicious participants with constant cost

Recall that with the assumption of at most $\frac{1}{3}$ participants are malicious, we have established the regret bound when the cost is constant. Without this assumption, we have proved the regret assuming distance-based cost, which highlights a gap. To this end, we next consider the constant cost that imposes more penalization and show the corresponding result. Intuitively, if the information from malicious participants is close enough to that from honest participants, the cost would always be constant, and thus the regret would be linear in $T$. As a result, we propose the following definition characterizing the difference between the two groups of participants and introduce an assumption accordingly, by generalizing Strictly Pre-fixed $\epsilon$-safe zone as in Appendix D.

**Pre-fixed $\epsilon$-safe zone**    A pre-fixed $\epsilon$-safe zone is defined as a set of participants $S_\epsilon$, such that for any participant $m \in S_\epsilon$ and any arm $1 \leq i \leq K$, we have that $f_i^m - h_i^m \geq \epsilon \cdot q_i^m$, where $f_i^j$ is the black-box reward generator, $h_i^j$ is the stochastic reward generator for arm $i$ with mean value $\mu_i$ with random seed $j$ and $q_i^j$ follows an unknown but fixed distribution different from that of $h_i^j$.

**Assumption 1.** *(Pre-fixed) The pre-fixed $\epsilon$-safe zone contains no malicious participants that only perform attacks on the estimators, namely, $M_A^1 \cap S_\epsilon = \emptyset$.*

We update the estimator computation, where the global estimator $\tilde{\mu}_i(t)$ is constructed as $\tilde{\mu}_i(t) = P_t \tilde{\mu}_i(t-1) + (1-P_t)\hat{\hat{\mu}}_i(t-1), \hat{\hat{\mu}}_i(t) = \frac{\sum_{m \in C_t^i} \bar{\mu}_i^m(t)}{|C_t^i|}$ with $C_t^i = \{1 \leq j \leq M : |\hat{\mu}_i(t) - \bar{\mu}_i^j(t)| \leq \frac{\epsilon}{2}\}$.

**Theorem 6.** *Assume the same conditions as in Theorem 4 except that the cost is constant. Let us further assume that Assumption 1 holds. With the new rule of updating the estimators, the regret bound of the proposed algorithm is $O(\log T)$.*

In Theorem 6 we have the same assumptions about honesty, and validators and commanders ($p_m$ and $w_m$ can be anything). The significant change here is constant cost.

### 4.2.4 General number of malicious participants with an efficient validator selection protocol

The requirement on honesty of validators lacks efficiency, taking significant time to achieve consensus. To address this, we propose a new approach to select validators based on a newly defined reputation score system motivated by the Proof-of-Authority concept (Fahim et al., 2023), allowing selecting any number of validators ranging from $M_H$ to $2M_H - 1$. More advantages are shown in Appendix F. More specifically, for each participant $i$, its reputation at time step $t$ is computed by a new smart contract as $RS_i^t = G(U_i^t)$, where $G$ is any monotonicity preserving function, and $U_i^t$ quantifies the accuracy of the information from participant $i$ at time step $t$, defined as $U_i^t = \sum_{j=1}^K -(\bar{\mu}_j^i(t-1) - \tilde{\mu}_j(t-1))^2 - \epsilon^2(\bar{\mu}_j^i(t-1) - \tilde{\mu}_j(t-1))^2)^2$ where $\bar{\bar{\mu}}_j^i(t-1)$ denotes the estimator for arm $j$ after the consensus step, and $\bar{\mu}_j^i(t-1), \tilde{\mu}_j(t-1)$ are the aforementioned estimators for arm $j$.

**Validator Selection**    First the protocol ranks the reputations of the participants and records the participants as $\{l_1, l_2, \ldots, l_M\}$ accordingly, where $l_1$ represents the participant with the largest reputation. Then the protocol selects the top $N$ participants, where $M_H \leq N \leq 2M_H - 1$.

**Theorem 7.** *Assume the same conditions as in Theorem 4 except that the cost is constant. Let us further assume that Assumption 1 holds, that the validators are selected based on Validator Selection, and there is at least one honest commander. Then we have that $E[R_T|A] \leq O(\log T)$.*

### 4.2.5 Other Material

Appendix A includes a discussion about other possible performance measures, Appendix B summaries of the contributions and future work, and Appendix C exhibits functions used in the model approach.

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

# A  OTHER PERFORMANCE MEASURE

While we have established various theoretical bounds on the regret of the coordination mechanism, demonstrating the algorithm's optimality, it is worth noting that security has been a crucial aspect of building fault-tolerant systems. In fact, we ensure that the security guarantee is necessary for the coordination mechanism's optimality, which is connected through our proposed framework as part of our contributions. In other words, security is an implication of the exhibited regret bounds. In the meantime, recall that to incentivize the participation of participants, we invented a new cost mechanism, motivated by (Murhekar et al., 2024). While our setting is not completely zero-sum, which does not enable the full characterization of Nash Equilibrium, the two different groups of participants, namely, malicious participants and honest participants, have conflicting objectives. To this end, we provide a qualitative discussion by illustrating the trade-offs faced by malicious participants and point out potential future directions regarding the cost mechanism. More specifically, we consider the following factors that affect security and illustrate how they are connected with regret.

## A.1  SECURITY OF THE PROTOCOL

**Digital signature**  The security of the coordination mechanism partially depends on the reliability of the signature scheme, as it determines whether a participant can maintain its own signature and the corresponding mapping between the label and signature. Note that the employment of the digital signature scheme (Goldwasser et al., 1988) is in a plug-in fashion, independent of everything else. As a result, the theoretical guarantee still holds, implying the security of the coordination mechanism and serving as a prerequisite needed for achieving consensus when running the Byzantine Fault Tolerant protocol.

**Consensus**  The security of the consensus protocol also plays an important role in the coordination mechanism's security, as no single participant can determine the estimator to be sent to the smart contract. This prevents malicious participants from manipulating the estimators but adds additional challenges for honest participants. By deploying the Byzantine Fault Tolerant protocol with the digital signature scheme and our newly proposed commander selection procedures, we guarantee that both consensus and good enough estimators are achieved with high probability. Only in this case can the regret be optimized, which implies that optimal regret indicates the security of the consensus protocol.

**Privacy**  Another main aspect of security is whether the participants' information is accessible to others, namely the degree of privacy preservation. We note that though the empirical reward estimators are available, the number of arm pulls is not broadcast. This prevents malicious participants from retrieving the reward and arm sequence, thus protecting privacy. Moreover, the rule for computing reputation is unknown to the participants, as it is implemented through a smart contract, which prevents malicious participants from manipulating the reputation. The correctness of the reputation is essential to the consensus protocol and thus the regret. In other words, the optimality of the regret also implies the correct execution of the reputation system.

## A.2  OPTIMALITY OF THE COST MECHANISM

This cost mechanism is consistent with the one in (Murhekar et al., 2024), by adding a cost term to the original reward. While their cost depends on how many samples a participant contributes, we measure how much contribution a participant makes to the validated estimators. Honest participants need to identify the malicious participants and gain knowledge about the reward to maximize their reward function.

Assuming the cost is constant, the optimal strategy for malicious participants is to send sufficiently accurate information so that the honest participants cannot determine their identities, which implies that there is no Nash Equilibrium. If the malicious participants keep broadcasting incorrect estimators, they would be excluded from consideration by the honest participants, allowing honest participants to incur a smaller cost. On the other hand, if the malicious participants send accurate enough information, the cost for honest participants is small as well, by definition. This implies that our proposed mechanism captures the trade-off and has the potential to uncover the Nash equilibrium with respect to how malicious participants transmit their estimators. We point out that quantitatively and rigorously characterizing the equilibrium presents a very promising direction, which goes beyond the scope of this paper.

## B  CONCLUSION AND FUTURE WORK

This paper considers a robust multi-agent multi-armed bandit (MAB) problem within the framework of system security, representing the first work to explore online sequential decision-making with participants distributed on a blockchain. The introduction of conditionally observable rewards and the penalization of inaccurate information brings new challenges, while taking security and privacy into consideration, besides optimality, distinct from blockchain-based federated learning or Byzantine-resilient multi-agent MAB. To solve the problem, we propose a new methodological approach combining the strategy based on Upper Confidence Bound (UCB) with blockchain techniques and invent new modifications. On a blockchain, a subset of participants forms a validator set responsible for information integration and achieving consensus on information transmitted by all participants. Consensus information is then sent to a smart contract for verification, with approved blocks only upon successful verification. The environment determines and sends the reward information to the participants based on the interaction with the smart contract. As part of our contributions, we use reputation to determine the validator selection procedure, which depends on the participants' historical behaviors. Additionally, we incorporate a digital signature scheme into the consensus process, eliminating the traditional $\frac{1}{3}$ assumption of the Byzantine general problem. Furthermore, we introduce a cost mechanism to incentivize malicious participants by rewarding their contributions to the verification step. We provide a comprehensive regret analysis demonstrating the optimality of our proposed algorithm under specific assumptions, marking a breakthrough in blockchain-related learning tasks, which has seen little analysis. To conclude, we also include a detailed discussion on the security and privacy guarantees.

While our framework works for a general number of malicious participants, it relies on assumptions about the structure of malicious behaviors. Removing such assumptions would generalize the problem setting. Meanwhile, we consider two types of attacks related to the framework—those targeting the estimators and those targeting the consensus—and note that there is a rich body of literature on different aspects of security attacks. Incorporating these into the framework is both meaningful and promising. Lastly, we emphasize that mechanism design has great potential in online learning, especially in a multi-agent system, to ensure that participants perform as expected. We hope that this work can pave the way for combining the rich literature in mechanism design with multi-agent learning systems, in the era of cybersecurity and mixed-motive cooperation.

## C  PSEUDO CODE OF SUB ALGORITHMS

---
**Algorithm 2** Validator or Commander Selection

---
1: **function** SELECTION($t, m, l, VRF$)
2:      Let $(pk_m, sk_m) = G(q_m)$
3:      Let $(hash, \pi) = VRF_F(\bar{q}_t, pk_m, sk_m)$
4:      $z = 0$
5:      **if** $\frac{hash}{2^{hl}} \notin [0, 1-l]$ **then**
6:          $z = 1$
7:      **end**
8:      If $\pi(hash, pk_m) = True$ then **return** $z$ else **return** 0
9: **end function**

---

## D  TERMINOLOGIES

**Existential Forgery**    Following the definition in (Goldwasser et al., 1988), malicious participants successfully perform an existential forgery if there exists a pair consisting of a message and a signature, such that the signature is produced by an honest participant.

**Adaptive Chosen Message Attack**    Consistent with (Goldwasser et al., 1988), we consider the most general form of a message attack, namely the adaptive chosen message attack. In this context, a malicious participant not only has access to the signatures of honest participants but also can determine what message to send after seeing these signatures. This grants the malicious participant a high degree of freedom, thereby making the attack more severe.

**Universal Composability Framework**    For homomorphic encryption, more specifically, secure multi-party computation, we follow the standard framework as in (Canetti, 2001). Specifically,

---

**Algorithm 3** Consensus

---

1: **function** CONSENSUS($S_C(t), \{B_t^m\}_m, M$)
2:     Run $sc_{sort}(S_V(t), PK)$ which returns sorted $S_C(t)$
3:     **for** $h = 1, 2, \ldots, |S_C(t)|$ **do**
4:         Generate the digital signature $\{s_h^m\}_m$ as in (Goldwasser et al., 1988)
5:         Define a message as $(s_h^m, B_t^m)$
6:         Execute Algorithm $SM(M)$ in (Lamport et al., 2019) with $S_C(t)[h]$ as the commander
7:         Derive the received information $\tilde{B}_t^h$ from $S_C(t)[h]$
8:         $v_t^m = 1$ if $\tilde{B}_t^h = B_t^m$ at honest participant $m$ and 0 otherwise
9:     **end**
10:     **if** $majority(v_t^m) = 1$ **then**
        Consensus is achieved and $B_t = B_t^m$
11:     **else**
        Consensus fails and $B_t = \emptyset$
12:     **return** $B_t$
13: **end function**

---

**Algorithm 4** Global Update

---

1: **function** GLOBAL_UPDATE($B_t$)
2:     **if** $B_t$ *is not empty* **then**
        Compute $\tilde{\mu}_i(t) = \frac{\sum_{m \in B_t} \bar{\mu}_i^m(t)}{|B_t|}$ for each $i \in \{1, \ldots, K\}$
3:     **else**
        $\tilde{\mu}_i(t) = \infty$ for each $i \in \{1, \ldots, K\}$
4:     **return** $(\tilde{\mu}_i(t))_{i \in \{1,\ldots,K\}}$
5: **end function**

---

**Algorithm 5** Operation

---

1: **function** OPERATION($\{\tilde{\mu}_i(t)\}_{i \in \{1,\ldots,K\}}, \{a_m^t\}_m, B_t, b_t$)
2:     Generate $r_{a_m^t}^m$ for every participant $m$
3:     **if** $b_t = 1$ *and* $B_t \subset M_H$ **then**
        Distribute $r_{a_m^t}^m$ and $\tilde{\mu}_i(t)$ for every $i$ to every participant $m$
4:     **if** $b_t = 1$ *and* $B_t \cap M_H < |B_t|$ **then**
        Distribute $r_{a_m^t}^m - c_t$ and $\tilde{\mu}_i(t)$ for every $i$ to every honest participant $m \in M_H$
        Distribute $r_{a_m^t}^m + c_t$ and $\tilde{\mu}_i(t)$ for every $i$ to every malicious participant $m \in M_A$
5:     **else**
        Distribute nothing to all participants
6:     **return**
7: **end function**

---

an exogenous environment, also known as an environment machine, interacts sequentially with a protocol. The process runs as follows. The environment sends some inputs to the protocol and receives outputs from the protocol that may contain malicious components. If there exists an ideal adversary such that the environment machine cannot distinguish the difference between interacting with this protocol or the ideal adversary, the protocol is deemed universally composable secure.

**Strict pre-fixed $\epsilon$-safe zone**  A pre-fixed $\epsilon, \delta$-safe zone is defined as a set of participants $S_\epsilon$, such that for any participant $j \in S_\epsilon$ and any arm $1 \le i \le K$, we have that the $f_i^j = (1 - \epsilon) \cdot h_i^j + \epsilon \cdot q_i^j$, where $f_i^j$ is the corresponding black-box reward generator, $h_i^j$ is the corresponding known stochastic reward generator for arm $i$ with mean value $\mu_i$ with random seed $j$ and $q_i^j$ follows an unknown but fixed distribution different from that of $h_i^j$.

This assumption separates the malicious participants from the honest participants to make the malicious participants distinguishable, thereby eliminating the estimators from malicious participants. It is worth noting that this assumption is consistent with the existing literature (Dubey and Pentland, 2020), which adopts the same principle when considering malicious behavior.

Moreover, this assumption can be relaxed to the version in our work where the minimum gap, instead of the exact gap, is $\epsilon$, which measures the difference between the estimators from the malicious participants and those from the honest participants.

# E  PROOF OF RESULTS IN SECTION 4

PROOF OF THEOREM 1

*Proof.* For regret, we have the following decomposition. Let us denote $b_t$ as the indicator function of whether the block at time step $t$ is approved. Likewise, for any time step $t$, we denote whether the estimators from the malicious participants are utilized in the integrated estimators as $h_t$. Let the length of the burn-in period be $L$.

Note that

$$
R_T = \max_i \sum_{m \in M_H} \sum_{t=1}^T \mu_i - \sum_{m \in M_H} \sum_{t=1}^T (\mu_{a_m^t}^b - c_t)
$$

$$
= \max_i \sum_{m \in M_H} \sum_{t=1}^T \mu_i - \sum_{m \in M_H} \sum_{t=1}^T \mu_{a_m^t}^b + \sum_{m \in M_H} \sum_{t=1}^T c_t
$$

$$
= \max_i \sum_{m \in M_H} \sum_{t=1}^T \mu_i - \sum_{m \in M_H} \sum_{t=1}^T \mu_{a_m^t} 1_{b_t=1} + \sum_{m \in M_H} \sum_{t=1}^T c_t
$$

$$
= \max_i \sum_{m \in M_H} \sum_{t=1}^T \mu_i - \sum_{m \in M_H} \sum_{t=1}^T \mu_{a_m^t} 1_{b_t=1} + \sum_{m \in M_H} \sum_{t=1}^T c 1_{h_t=1}
$$

Meanwhile, the regret can be bounded as follows

$$
R_T \le L + c \cdot L + \sum_{t=L+1}^T \sum_{m \in M_H} (\mu_{i^*} - \mu_{a_m^t} 1_{b_t=1}) + \sum_{m \in M_H} \sum_{t=L+1}^T c 1_{h_t=1}
$$

$$
\doteq (c + 1) \cdot L + T_1 + T_2 \tag{2}
$$

We start with the second term $T_2$. Note that $h_t = 1$ is equivalent to $\{m : m \in B_t \cap m \notin M_H\} \ne \emptyset$. Note that because the cost is positive, $B_t$ is nonempty.

By taking the expectation over $T_2$, we derive

$$E[T_2|A] = \sum_{m \in M_H} \sum_{t=L+1}^{T} cE[1_{h_t=1}]$$

$$= \sum_{m \in M_H} \sum_{t=L+1}^{T} cE[1_{\{m:m \in B_t \cap m \notin M_H\} \neq \emptyset}]$$

Based on Lemma 2 in (Zhu et al., 2023), we obtain that

$$1_{\{m:m \in B_t \cap m \notin M_H\} \neq \emptyset} = 1_{|A_t| < 2f}$$

which immediately implies that

$$E[T_2|A] = \sum_{m \in M_H} \sum_{t=L+1}^{T} cE[1_{h_t=1}]$$

$$= \sum_{m \in M_H} \sum_{t=L+1}^{T} cE[1_{\{m:m \in B_t \cap m \notin M_H\} \neq \emptyset}]$$

$$= \sum_{m \in M_H} \sum_{t=L+1}^{T} cE[1_{|A_t| < 2f}].$$

In the meantime, we note that for any honest validators, the choice of $A_t$ guarantees that honest participants are included after the burn-in period. More specifically, the set of $A_t$ satisfies that for any validator $j \in M_H$,

$$m \in A_t \Leftrightarrow k_i n_{m,i}(t) > n_{j,i}(t) \Leftrightarrow m \in M_H$$

where $1 < k_i < 2$. This condition holds at the end of burn-in period which is straightforward since each honest. After the burn-in period, the honest participants has the same decision rule

$$a_m^t = argmax_i \tilde{\mu}_i^m(t) + F(m,i,t)$$

where $\tilde{\mu}_i^m(t) = \tilde{\mu}_i^b(t)$. In other words, each honest participant uses the validated estimator $\tilde{\mu}_i^b(t)$. Since both $n_{m,i}(t)$ and $n_{j,i}(t)$ are larger than $\frac{L}{K}$, then we have that there exists $k_i = \frac{n_{j,i}(t)K}{L}$, such as $k_i n_{m,i}(t) > n_{j,i}(t)$ for every $m \in M_H$.

This implies that

$$A_t > |M_H| \geq 2f \tag{3}$$

by the assumption that the number of honest participants is at least $\frac{2}{3}M$.

That is to say,

$$E[1_{|A_t| > 2f}] = 1$$

and subsequently, we have

$$E[T_2|A] = \sum_{m \in M_H} \sum_{t=L+1}^{T} cE[1_{|A_t| < 2f}]$$

$$= 0$$

We note that the construction of $A_t$ is done without knowing the number of pulls of arms of other participants. This is realized by using the homomorphic results, Theorem 5.2 as in (Asharov et al., 2012) under the universal composability framework.

Next, we proceed to bound the first term $T_1$. Note that

$$E[T_1|A] \leq \sum_{t=L+1}^{T} \sum_{m \in M_H} (\mu_{i^*} - \mu_{a_m^t} 1_{b_t=1})$$

$$= (T-L) \cdot |M_H| \cdot \mu_{i^*} - \sum_{m \in M_H} \sum_{t=L}^{T} E[\mu_{a_m^t}|b_t=1] P(b_t=1)$$

In the meantime, we obtain the following

$$E[\mu_{a_m^t}|b_t=1]$$

$$= E[\sum_{k=1}^{K} \mu_k \cdot 1_{a_m^t=k}|b_t=1]$$

$$= \sum_{k=1}^{K} E[\mu_k 1_{a_m^t=k}|b_t=1]$$

$$\geq \sum_{k=1}^{K} \mu_k \cdot \frac{1}{P(b_t=1)} \cdot (E[1_{a_m^t=k}] - P(b_t=0)).$$

This immediately gives us that

$$E[T_1|A]$$

$$\leq (T-L)|M_H|\mu_{i^*} - \sum_{m \in M_H} \sum_{t=L}^{T} \sum_{k=1}^{K} (\sum \mu_k \cdot \frac{1}{P(b_t=1)} \cdot (E[1_{a_m^t=k}] - P(b_t=0)))P(b_t=1)$$

$$= (T-L)|M_H|\mu_{i^*} - \sum_{m \in M_H} \sum_{t=L}^{T} \sum_{k=1}^{K} (\sum \mu_k (E[1_{a_m^t=k}] - P(b_t=0)))$$

$$= (T-L)|M_H|\mu_{i^*} - \sum_{m \in M_H} \sum_{t=L}^{T} \sum_{k=1}^{K} \mu_k E[1_{a_m^t=k}] + \sum_{m \in M_H} \sum_{t=L}^{T} \sum_{k=1}^{K} \mu_k P(b_t=0). \quad (4)$$

Based on Theorem 2 in (Lamport et al., 2019), the consensus is achieved, i.e. $b_t = 1$, as long as the digital signatures of the honest participants can not be forged. Based on our assumption, we have that the malicious participants can only perform existential forgery on the signatures of the honest participants and the attacks are adaptive chosen-message attack. Then based on the result, Main Theorem in (Goldwasser et al., 1988), the attack holds with probability at most $\frac{1}{Q(l)}$ for any polynomial function $Q$ and large enough $l$ where $l$ is the length of the signature.

More precisely, we have that with probability at least $1 - \frac{1}{Tl^{T-1}}$, the signature of the honest participants can not be forged, and thus, the consensus can be achieved, i.e.

$$P(b_t=1) \geq 1 - \frac{1}{Tl^{T-1}}. \quad (5)$$

Subsequently, we derive that

$$(14) \leq (T-L)|M_H|\mu_{i^*} - \sum_{m \in M_H} \sum_{t=L}^{T} \sum_{k=1}^{K} \mu_k E[1_{a_m^t=k}] + \sum_{m \in M_H} \sum_{t=L}^{T} \sum_{k=1}^{K} \mu_k (\frac{1}{Tl^{T-1}})$$

$$\leq \sum_{m \in M_H} \sum_{t=L}^{T} (\mu_{i^*} - \sum_{k=1}^{K} \mu_k E[1_{a_m^t=k}]) + |M_H|Kl^{T-1}$$

$$= \sum_{m \in M_H} \sum_{k=1}^{K} \Delta_k E[n_{m,k}(t)] + |M_H|Kl^{T-1}$$

$$\doteq T_{21} + |M_H|Kl^{T-1}$$

And for each honest participant, they are using the estimators based on the validated estimators, as long as the block is approved. Consider the following event, $A = \{\forall 1 \leq t \leq T, b_t = 1\}$. Based on (15) and the Bonferroni's inequality, we obtained that

$$
\begin{aligned}
P(A) &= P(\forall 1 \leq t \leq T, b_t = 1) \\
&= 1 - P(\exists 1 \leq t \leq T, b_t = 0) \\
&\geq 1 - \sum_{t=1}^{T} P(b_t = 0) \\
&\geq 1 - \frac{1}{l^{T-1}}.
\end{aligned}
$$

On event $A$, the blockchain always gets approved, and then all the honest participants follow the validated estimators from the validators. By (3) and Lemma 2 in (Zhu et al., 2023), we have that the validated estimator $\tilde{\mu}_i(t)$ can be expressed as

$$
\hat{\mu}_i(t) = \sum_{j \in A_t \cap M_H} w_{j,i}(t) \bar{\mu}_i^j(t)
$$

where the weight $w_{j,i}(t)$ meets the condition

$$
\sum_{j \in A_t \cap M_H} w_{j,i}(t) = 1,
$$

which immediately implies that

$$
E[\hat{\mu}_i(t)] = \mu_i.
$$

We note that the variance of $\hat{\mu}_i(t)$, $var(\hat{\mu}_i(t))$, satisfies that,

$$
\begin{aligned}
var(\hat{\mu}_i(t)) &= var\left( \sum_{j \in A_t \cap M_H} w_{j,i}(t) \bar{\mu}_i^j(t) \right) \\
&\leq |A_t \cap M_H| \sum_{j \in A_t \cap M_H} w_{j,i}(t)^2 var(\bar{\mu}_i^j(t))) \\
&\leq |A_t \cap M_H| \sum_{j \in A_t \cap M_H} w_{j,i}^2(t) \sigma^2 \frac{1}{n_{j,i}(t)} \\
&\leq |A_t \cap M_H| \sum_{j \in A_t \cap M_H} w_{j,i}^2(t) \sigma^2 \frac{k_i}{n_{m,i}(t)} \\
&= |M_H| \frac{k_i}{n_{m,i}(t)} \sum_{j \in M_H} w_{j,i}^2(t) \sigma^2 \\
&\leq |M_H| \frac{k_i \sigma^2}{n_{m,i}(t)}
\end{aligned}
$$

where the inequality holds by the Cauchy-Schwarz inequality, the second inequality holds by the definition of sub-Gaussian distributions, the third inequality results from the construction of $A_t$, and the last inequality is as a result of $(a + b)^2 \geq a^2 + b^2$.

Next, we show by induction that $var(\tilde{\mu}_i(t)) \leq 3|M_H| \frac{k_i \sigma^2}{n_{m,i}(t)}$ for $t \geq 3K$.

At time step $3K$, we have that $var(\tilde{\mu}_i(t)) \leq 1$ since $E[\tilde{\mu}_i(t)] = \mu_i \leq 1$. In the meantime,

$$
\begin{aligned}
3|M_H| &\frac{k_i \sigma^2}{n_{m,i}(t-1)} \\
&\geq 3|M_H| \frac{k_i \sigma^2}{3} \\
&= |M_H| k_i \sigma^2 \geq 1
\end{aligned}
$$

since we have $k_i \geq 1$ and $\sigma^2 \geq \frac{1}{M_H}$.

First, assume that for $t - 1$, we have $var(\tilde{\mu}_i(t-1)) \leq 3|M_H|\frac{k_i\sigma^2}{n_{m,i}(t-1)}$.

Meanwhile, by the update rule such that $\tilde{\mu}_i(t) = (1 - P_t)\hat{\mu}_i(t) + P_t\tilde{\mu}_i(\tau)$ where $\tau = \max_{s<t}\{b_s = 1\}$.

Note that with probability at least $P(A) = 1 - \frac{1}{l^{T-1}}$, $b_s = 1$ for all $s < t$. This implies that on event $A$, $\tau = t - 1$. Therefore, by the cauchy-schwartz inequality, we obtain that

$$var(\tilde{\mu}_i(t)) \leq 2(1 - P_t)^2(var(\hat{\mu}_i(t))) + 2P_t^2 var(\tilde{\mu}_i(t-1))$$
$$\leq \frac{1}{2}|M_H|\frac{k_i\sigma^2}{n_{m,i}(t)} + \frac{1}{2}3|M_H|\frac{k_i\sigma^2}{n_{m,i}(t-1)}$$
$$\leq 3|M_H|\frac{k_i\sigma^2}{n_{m,i}(t)}$$

where the last inequality holds by the fact that $n_{m,i}(t-1) \geq n_{m,i}(t) - 1 \geq \frac{2}{3}n_{m,i}(t)$ when $t > 3 \cdot K$.

Subsequently, we have that

$$P(\tilde{\mu}_i^m(t) - \sqrt{\frac{C_1 \log t}{n_{m,i}(t)}} > \mu_i, n_{m,i}(t-1) \geq l)$$
$$\leq \exp\{-\frac{(\sqrt{\frac{C_1 \log t}{n_{m,i}(t)}})^2}{2var(\tilde{\mu}_i^m)}\}$$
$$\leq \exp\{-\frac{(\sqrt{\frac{C_1 \log t}{n_{m,i}(t)}})^2}{6|M_H|\frac{k_i\sigma^2}{n_{m,i}(t)}}\}$$
$$= \exp\{-\frac{C_1 \log t}{6|M_H|k_i\sigma^2}\} \leq \frac{1}{t^2} \tag{6}$$

where the first inequality holds by Chernoff bound, the second inequality is derived by plugging in the above upper bound on $var(\tilde{\mu}_i^m(t))$, and the last inequality results from then choice of $C_1$ that satisfies $\frac{C_1}{6|M_H|k_i\sigma^2} \geq 1$.

Likewise, by symmetry, we have

$$P(\tilde{\mu}_i^m(t) + \sqrt{\frac{C_1 \log t}{n_{m,i}(t)}} < \mu_i, n_{m,i}(t-1) \geq l) \leq \frac{1}{t^2}. \tag{7}$$

Meanwhile, we have that

$$\sum_{t=L+1}^{T} P(\mu_i + 2\sqrt{\frac{C_1 \log t}{n_{m,i}(t-1)}} > \mu_{i^*}, n_{m,i}(t-1) \geq l) = 0 \tag{8}$$

if the choice of $l$ satisfies $l \geq \lceil\frac{4C_1 \log T}{\Delta_i^2}\rceil$ with $\Delta_i = \mu_{i^*} - \mu_i$.

Based on the decision rule, we have the following hold for $n_{m,i}(T)$ with $l \geq \lceil \frac{4C_1 \log T}{\Delta_i^2} \rceil$,

$$n_{m,i}(T) \leq l + \sum_{t=L+1}^{T} 1_{\{a_t^m = i, n_{m,i}(t) > l\}}$$

$$\leq l + \sum_{t=L+1}^{T} 1_{\{\tilde{\mu}_i^m - \sqrt{\frac{C_1 \log t}{n_{m,i}(t-1)}} > \mu_i, n_{m,i}(t-1) \geq l\}}$$

$$+ \sum_{t=L+1}^{T} 1_{\{\tilde{\mu}_{i*}^m + \sqrt{\frac{C_1 \log t}{n_{m,i*}(t-1)}} < \mu_{i*}, n_{m,i}(t-1) \geq l\}}$$

$$+ \sum_{t=L+1}^{T} 1_{\{\mu_i + 2\sqrt{\frac{C_1 \log t}{n_{m,i}(t-1)}} > \mu_{i*}, n_{m,i}(t-1) \geq l\}}.$$

By taking the expectation over $n_{m,i}(t)$, we obtain

$$E[n_{m,i}(t)] \leq l + \sum_{t=L+1}^{T} P(\tilde{\mu}_i^m(t) - \sqrt{\frac{C_1 \log t}{n_{m,i}(t)}} > \mu_i, n_{m,i}(t-1) \geq l)$$

$$+ \sum_{t=L+1}^{T} P(\tilde{\mu}_i^m(t) + \sqrt{\frac{C_1 \log t}{n_{m,i}(t)}} < \mu_i, n_{m,i}(t-1) \geq l)$$

$$+ \sum_{t=L+1}^{T} P(\mu_i + 2\sqrt{\frac{C_1 \log t}{n_{m,i}(t-1)}} > \mu_{i*}, n_{m,i}(t-1) \geq l)$$

$$\leq l + \sum_{t=L+1}^{T} \frac{1}{t^2} + \sum_{t=L+1}^{T} \frac{1}{t^2} + 0$$

$$\leq l + \frac{\pi^2}{3} = \lceil \frac{4C_1 \log T}{\Delta_i^2} \rceil + \frac{\pi^2}{3} \tag{9}$$

where the second inequality holds by using (6), (7), and (18).

Then by the definition of $T_{21}$, we derive

$$E[T_{21}|A] = \sum_{m \in M_H} \sum_{k=1}^{K} \Delta_k E[n_{m,k}(t)]$$

$$\leq \sum_{m \in M_H} \sum_{k=1}^{K} \Delta_k (\lceil \frac{4C_1 \log T}{\Delta_i^2} \rceil + \frac{\pi^2}{3})$$

where the inequality results from (17).

Consequently, we obtain

$$(14) \leq E[T_{21}|A] + |M_H| K l^{T-1}$$

$$\leq \sum_{m \in M_H} \sum_{k=1}^{K} \Delta_k (\lceil \frac{4C_1 \log T}{\Delta_i^2} \rceil + \frac{\pi^2}{3}) + |M_H| K l^{T-1}. \tag{10}$$

Furthermore, we have

$$(23) \leq (c+1) \cdot L + E[T_1|A] + E[T_2|A]$$

$$\leq (c+1) \cdot L + \sum_{m \in M_H} \sum_{k=1}^{K} \Delta_k (\lceil \frac{4C_1 \log T}{\Delta_i^2} \rceil + \frac{\pi^2}{3}) + |M_H| K l^{T-1} + 0 \tag{11}$$

which completes the proof.

$\square$

PROOF OF THEOREM 2

*Proof.* By the same deifnition of the regret, we, again, have the following regret decomposition
Note that

$$R_T = \max_i \sum_{m \in M_H} \sum_{t=1}^{T} \mu_i - \sum_{m \in M_H} \sum_{t=1}^{T} (\mu_{a_m^t}^b - c_t)$$

$$= \max_i \sum_{m \in M_H} \sum_{t=1}^{T} \mu_i - \sum_{m \in M_H} \sum_{t=1}^{T} \mu_{a_m^t}^b + \sum_{m \in M_H} \sum_{t=1}^{T} c_t$$

$$= \max_i \sum_{m \in M_H} \sum_{t=1}^{T} \mu_i - \sum_{m \in M_H} \sum_{t=1}^{T} \mu_{a_m^t} 1_{b_t=1} + \sum_{m \in M_H} \sum_{t=1}^{T} c_t$$

$$= \max_i \sum_{m \in M_H} \sum_{t=1}^{T} \mu_i - \sum_{m \in M_H} \sum_{t=1}^{T} \mu_{a_m^t} 1_{b_t=1} + \sum_{m \in M_H} \sum_{t=1}^{T} c_t 1_{h_t=1}$$

Meanwhile, the regret can be bounded as follows

$$R_T \le L + c \cdot L + \sum_{t=L+1}^{T} \sum_{m \in M_H} (\mu_{i^*} - \mu_{a_m^t} 1_{b_t=1}) + \sum_{m \in M_H} \sum_{t=L+1}^{T} c_t 1_{h_t=1}$$

$$\doteq (c+1) \cdot L + T_1 + T_2 \tag{12}$$

We start with the second term $T_2$. Note that $h_t = 1$ is equivalent to $\{m : m \in B_t \cap m \notin M_H\} \neq \emptyset$.
By taking the expectation over $T_2$, we derive

$$E[T_2|A] = \sum_{m \in M_H} \sum_{t=L+1}^{T} E[c_t \cdot 1_{h_t=1}]$$

$$= \sum_{m \in M_H} \sum_{t=L+1}^{T} E[c_t \cdot 1_{\{m:m \in B_t \cap m \notin M_H\} \neq \emptyset}]$$

By the Chernoff-Hoeffding's inequality and choosing $\eta_t \ge \frac{\sqrt{\log t}}{\sqrt{n_i(t)}}$, we obtain that

$$P(|\bar{\mu}_i^m(t) - \mu_i| \ge \eta_t)$$

$$= P(|\bar{\mu}_i^m(t) - \mu_i| \ge \frac{\sqrt{\log t}}{\sqrt{n_i(t)}})$$

$$\le 2 \exp\{-\frac{\log t}{4\sigma^2 n_{m,i}^2(t)}\}$$

$$= 2 \exp\{-\frac{\log t}{4\sigma^2 n_{m,i}^2(t)}\}$$

$$\le \frac{1}{t^2} \doteq P_t,$$

when $t > L$, i.e. after the burn-in period.

If $c_t \le \frac{1}{t}$, then we have that $E[T_2|A] \le \log T$, which presents an upper bound on $T_2$.

If $c_t = Dist(\tilde{\mu}_i(t), \mu_i)$, then based on the definition of $B_t$ and $m \in B_t$ as in Option 2, we have that
$\bar{\mu}_i^m(t)$ is smaller than the top $f$ values and larger than the below $f$ values. Based on Theorem 1 as in
(Dong et al., 2023), we have that

$$||\hat{\mu}_i(t) - \bar{z}_i(t)|| \le c_\delta \Delta^2$$

where $\Delta$ represents the largest distance between the honest estimators and $\bar{z}_i(t)$ that is the averaged estimator maintained by all the honest participants.

Then by definition, we obtain that

$$\Delta = \max_{i \in M_H} |\bar{\mu}_i(t) - \bar{z}_i(t)|$$

$$\leq \max_{i,j \in M_H} [|\bar{\mu}_i(t) - \mu_i| + |\bar{\mu}_j(t) - \mu_i|]$$

$$\leq 2\eta_t$$

which holds with probability $1 - P_t$.

Therefore, we have that with probability $1 - P_t$

$$|\hat{\mu}_i(t) - \bar{z}_i(t)| \leq 2c_\delta \eta_t$$

and

$$|\bar{z}_i(t) - \mu_i| \leq \eta_t$$

which holds by the Chernoff Bound inequality.

Subsequently, we obtain that with probability $1 - P_t$

$$|\hat{\mu} - \mu_i| \leq |\hat{\mu}_i(t) - \bar{z}_i(t)| + |\bar{z}_i(t) - \mu_i|$$

$$\leq (2c_\delta + 1)\eta_t^6$$

Meanwhile, for the distance measure, we have with probability $1 - P_t$

$$Dist(\tilde{\mu}_i(t) - \mu_i) = |\tilde{\mu}_i(t) - \mu_i|^6$$

$$= |\bar{q}_t \tilde{\mu}_i(t-1) + (1 - \bar{q}_t)\hat{\mu}_i(t) - \mu_i|^6$$

$$\leq \bar{q}_t |\tilde{\mu}_i(t-1) - \mu_i|^6 + (1 - \bar{q}_t)|\hat{\mu}_i(t) - \mu_i|^6$$

$$\leq \bar{q}_t Dist(\tilde{\mu}_i(t-1), \mu_i) + (1 - \bar{q}_t)(2c_\delta + 1)^6 \eta_t^6 \quad (13)$$

Since by definition, we derive that

$$P(Dist(\tilde{\mu}_i(L), \mu_i) \geq O(\frac{\eta_t^2}{n_i(t)}))$$

$$\leq P(Dist(\tilde{\mu}_i(L), \mu_i) \geq O(\frac{\log t^3}{n_i(t)^3}))$$

$$\leq P(|\tilde{\mu}_i(L) - \mu_i| \geq O(\frac{\sqrt{\log t}}{\sqrt{n_i(t)}}))$$

$$\leq P(|\tilde{\mu}_i(L) - \mu_i| \geq \eta_t)$$

$$= P_t$$

That is to say, with probability $1 - P_t$,

$$Dist(\tilde{\mu}_i(L), \mu_i) \leq O(\frac{\eta_L^2}{n_i(L)})$$

Next, suppose that at each time step $t$, with probability $1 - P_t$, $Dist(\tilde{\mu}_i(t), \mu_i) \leq O(\frac{\eta_t^2}{n_i(t)})$.

Then by choosing $\bar{q}_t = 1 - \frac{1}{n_i(t)}$ and 13, we have that

$$Dist(\tilde{\mu}_i(t+1), \mu_i)$$

$$\leq \bar{q}_t Dist(\tilde{\mu}_i(t), \mu_i) + (1 - \bar{q}_t)(2c_\delta + 1)^6 \eta_t^6$$

$$\leq O(\frac{\eta_t^2}{n_i(t)}) + O(\frac{1}{n_i(t)}\eta_t^6)$$

$$= O(\frac{\eta_{t+1}^2}{n_i(t+1)})$$

Then we use the mathematical induction and derive that for any $t \geq L$, with probability $1 - P_t$,

$$Dist(\tilde{\mu}_i(t), \mu_i) \leq O(\frac{\eta_t^2}{n_i(t)}).$$

By the definition of cost, we obtain that with probability $1 - P_t$

$$c_t = \min_i Dist(\tilde{\mu}_i(t), \mu_i)$$

$$\leq O(\frac{\log t}{\max_i n_i(t)^2}) = O(\frac{\log t}{t^2})$$

where the last inequality holds by the fact that $\max_i n_i(t) \geq \frac{\sum_i n_i(t)}{K} = O(t)$.

Then we drive that

$$E[T_2|A] \leq \sum_{m \in M_H} \sum_{t=L+1}^{T} E[c_t \cdot 1_{\{m:m \in B_t \cap m \notin M_H\} \neq \emptyset}]$$

$$\leq \sum_{m \in M_H} \sum_{t=L+1}^{T} E[c_t]$$

$$\leq \sum_{m \in M_H} \sum_{t=L+1}^{T} [(1 - P_t) \cdot O(\frac{\log t}{t^2}) + P_t]$$

$$= \sum_{m \in M_H} \sum_{t=L+1}^{T} O(\frac{\log t}{t^2})$$

$$\leq \log T \sum_{m \in M_H} \sum_{t=L+1}^{T} O(\frac{1}{t^2}) = O(\log T).$$

We next follow the same steps as in the proof of Theorem 1 for bounding $E[T_1]$. Note that

$$E[T_1|A] \leq \sum_{t=L+1}^{T} \sum_{m \in M_H} (\mu_{i^*} - \mu_{a_m^t} 1_{b_t=1})$$

$$= (T - L) \cdot |M_H| \cdot \mu_{i^*} - \sum_{m \in M_H} \sum_{t=L}^{T} E[\mu_{a_m^t}|b_t = 1]P(b_t = 1)$$

In the meantime, we obtain the following

$$E[\mu_{a_m^t}|b_t = 1]$$

$$= E[\sum_{k=1}^{K} \mu_k \cdot 1_{a_m^t=k}|b_t = 1]$$

$$= \sum_{k=1}^{K} E[\mu_k 1_{a_m^t=k}|b_t = 1]$$

$$\geq \sum_{k=1}^{K} \mu_k \cdot \frac{1}{P(b_t = 1)} \cdot (E[1_{a_m^t=k}] - P(b_t = 0)).$$

This immediately gives us that

$$E[T_1|A]$$

$$\leq (T-L)|M_H|\mu_{i^*} - \sum_{m \in M_H} \sum_{t=L}^{T} \sum_{k=1}^{K} (\sum \mu_k \cdot \frac{1}{P(b_t=1)} \cdot (E[1_{a_m^t=k}] - P(b_t=0)))P(b_t=1)$$

$$= (T-L)|M_H|\mu_{i^*} - \sum_{m \in M_H} \sum_{t=L}^{T} \sum_{k=1}^{K} (\sum \mu_k (E[1_{a_m^t=k}] - P(b_t=0))$$

$$= (T-L)|M_H|\mu_{i^*} - \sum_{m \in M_H} \sum_{t=L}^{T} \sum_{k=1}^{K} \mu_k E[1_{a_m^t=k}] + \sum_{m \in M_H} \sum_{t=L}^{T} \sum_{k=1}^{K} \mu_k P(b_t=0). \tag{14}$$

Based on Theorem 2 in (Lamport et al., 2019), the consensus is achieved, i.e. $b_t = 1$, as long as the digital signatures of the honest participants can not be forged. Based on our assumption, we have that the malicious participants can only perform existential forgery on the signatures of the honest participants and the attacks are adaptive chosen-message attack. Then based on the result, Main Theorem in (Goldwasser et al., 1988), the attack holds with probability at most $\frac{1}{Q(l)}$ for any polynomial function $Q$ and large enough $l$ where $l$ is the length of the signature.

More precisely, we have that with probability at least $1 - \frac{1}{l^T}$, the signature of the honest participants can not be forged, and thus, the consensus can be achieved, i.e.

$$P(b_t=1) \geq 1 - \frac{1}{l^T}. \tag{15}$$

Consequently, we have

$$E[T_1] \leq (T-L)|M_H|\mu_{i^*} - \sum_{m \in M_H} \sum_{t=L}^{T} \sum_{k=1}^{K} \mu_k E[1_{a_m^t=k}] + \sum_{m \in M_H} \sum_{t=L}^{T} \sum_{k=1}^{K} \mu_k (\frac{1}{l^T})$$

$$\leq \sum_{m \in M_H} \sum_{t=L}^{T} (\mu_{i^*} - \sum_{k=1}^{K} \mu_k E[1_{a_m^t=k}]) + |M_H|Kl^{T-1}$$

$$= \sum_{m \in M_H} \sum_{k=1}^{K} \Delta_k E[n_{m,k}(t)] + |M_H|Kl^{T-1} \tag{16}$$

Based on the decision rule, we have the following hold for $n_{m,i}(T)$ with $l \geq \lceil \frac{4C_1 \log T}{\Delta_i^2} \rceil$,

$$n_{m,i}(T) \leq l + \sum_{t=L+1}^{T} 1_{\{a_t^m=i, n_{m,i}(t)>l\}}$$

$$\leq l + \sum_{t=L+1}^{T} 1_{\{\tilde{\mu}_i^m - \sqrt{\frac{C_1 \log t}{n_{m,i}(t-1)}} > \mu_i, n_{m,i}(t-1) \geq l\}}$$

$$+ \sum_{t=L+1}^{T} 1_{\{\tilde{\mu}_{i^*}^m + \sqrt{\frac{C_1 \log t}{n_{m,i^*}(t-1)}} < \mu_{i^*}, n_{m,i}(t-1) \geq l\}}$$

$$+ \sum_{t=L+1}^{T} 1_{\{\mu_i + 2\sqrt{\frac{C_1 \log t}{n_{m,i}(t-1)}} > \mu_{i^*}, n_{m,i}(t-1) \geq l\}}.$$

By taking the expectation over $n_{m,i}(t)$, we obtain

$$E[n_{m,i}(t)] \leq l + \sum_{t=L+1}^{T} P(\tilde{\mu}_i^m(t) - \sqrt{\frac{C_1 \log t}{n_{m,i}(t)}} > \mu_i, n_{m,i}(t-1) \geq l)$$

$$+ \sum_{t=L+1}^{T} P(\tilde{\mu}_i^m(t) + \sqrt{\frac{C_1 \log t}{n_{m,i}(t)}} < \mu_i, n_{m,i}(t-1) \geq l)$$

$$+ \sum_{t=L+1}^{T} P(\mu_i + 2\sqrt{\frac{C_1 \log t}{n_{m,i}(t-1)}} > \mu_{i^*}, n_{m,i}(t-1) \geq l) \qquad (17)$$

Recall that by our concentration inequality, we obtain that

$$P(\tilde{\mu}_i^m(t) + (\frac{C_1 \log t}{n_{m,i}(t)})^{\frac{1}{6}} < \mu_i, n_{m,i}(t-1) \geq l)$$

$$\leq P(|\tilde{\mu}_i^m(t) - \mu_i| \geq O(\frac{\log t^{\frac{1}{6}}}{n_i(t)^{\frac{1}{3}}}), n_{m,i}(t-1) \geq l)$$

$$= P(Dist(\tilde{\mu}_i^m(t), \mu_i) \geq O(\frac{\eta_t^2}{n_i(t)}), n_{m,i}(t-1) \geq l)$$

$$\leq P_t = \frac{1}{t^2}.$$

Meanwhile, we have that

$$\sum_{t=L+1}^{T} P(\mu_i + 2(\frac{C_1 \log t}{n_{m,i}(t-1)})^{\frac{1}{6}} > \mu_{i^*}, n_{m,i}(t-1) \geq l) = 0 \qquad (18)$$

if the choice of $l$ satisfies $l \geq \lceil \frac{4C_1 \log T}{\Delta_i^6} \rceil$ with $\Delta_i = \mu_{i^*} - \mu_i$.

This immediately implies that

$$E[n_{m,i}(t)] \leq l + \sum_{t=L+1}^{T} P_t + \sum_{t=L+1}^{T} P_t + 0$$

$$\leq l + \frac{\pi^2}{3}$$

$$= O(\log T).$$

Then, by 16, we arrive at

$$E[T_1] \leq \sum_{m \in M_H} \sum_{k=1}^{K} \Delta_k E[n_{m,k}(t)] + |M_H| K l^{T-1}$$

$$\leq O(\log T) + |M_H| K l^{T-1}$$

Henceforth, based on 23, we have the following upper bound on the regret

$$E[R_T|A] \leq (c+1) \cdot L + E[T_1|A] + E[T_2|A]$$

$$\leq (c+1) \cdot L + O(\log T) + |M_H| K l^{T-1} \qquad (19)$$

which completes the proof.

PROOF OF THEOREM 3

*Proof.* Again, we start by decomposing the regret as

$$R_T \leq L + c \cdot L + \sum_{t=L+1}^{T} \sum_{m \in M_H} (\mu_{i^*} - \mu_{a_m^t} 1_{b_t=1}) + \sum_{m \in M_H} \sum_{t=L+1}^{T} c_t 1_{h_t=1}$$

$$\doteq (c+1) \cdot L + T_1 + T_2$$

We note that the consensus protocol runs $M$ times, with each validator (i.e., participant in this case) being selected as a commander. For any malicious participant $j \in M_A^2$, it serves as a commander and is thus included in $D_t$. This holds true because, according to Lemma 3 in (Goldwasser et al., 1988), if the message is a chandelier tree generated by the secret key $SK_m$ of participant $m$, any participant can verify the public key $PK_m$, or equivalently, trace back to the root of the signature tree of the message sender. Due to the unique mapping between $PK_m$ and $m$, the honest participant keeps a record of the vertex index of the malicious participants that attack the consensus.

This implies that $j \notin B_t$, i.e. the set $B_t$ can only contain estimators from either honest participants or set $M_A^1$ that satisfies $|M_A^1| < M_H - 1$. Therefore, the property of $B_t$ follows from that as in Option 2, which essentially indicates that Option 3 is equivalent to Option 2 with at least one half honest participants. Considering that the remaining algorithmic steps are the same, the analysis of $T_1$ and $T_2$ is consistent with that of Theorem 2.

Consequently, we have that

$$E[T_1] \leq \sum_{m \in M_H} \sum_{k=1}^{K} \Delta_k E[n_{m,k}(t)] + |M_H| K l^{T-1}$$

$$\leq O(\log T) + |M_H| K l^{T-1}$$

and

$$E[T_2] \leq \log T \sum_{m \in M_H} \sum_{t=L+1}^{T} O(\frac{1}{t^2}) = O(\log T).$$

Subsequently, we derive the same regret bound as in Theorem, as

$$E[R_T] \leq (c+1) \cdot L + O(\log T) + |M_H| K l^{T-1} + \log T \sum_{m \in M_H} \sum_{t=L+1}^{T} O(\frac{1}{t^2})$$

$$= O(\log T)$$

which completes the proof.

$\square$

PROOF OF THEOREM 4

*Proof.* The proof of Theorem 4 is similar to that of Theorem 3 as follows. The regret of the coordination mechanism is again decomposed as

$$R_T \leq L + c \cdot L + \sum_{t=L+1}^{T} \sum_{m \in M_H} (\mu_{i^*} - \mu_{a_m^t} 1_{b_t=1}) + \sum_{m \in M_H} \sum_{t=L+1}^{T} c_t 1_{h_t=1}$$

$$\doteq (c+1) \cdot L + T_1 + T_2$$

For malicious participant $j \in M_A^2$, it only attacks the consensus process and does not attack the estimators. In the meantime, for malicious participant $l \in M_A^1$, it only attacks the estimators, but does not attack the consensus process. Since $|M_A^1| < \frac{1}{2}M - 1$, when using Option 2, the set $B_t$ is the same as the case where only at most $\frac{1}{2}M$ participants are malicious. Therefore, we have that

$$E[T_2] \leq \log T \sum_{m \in M_H} \sum_{t=L+1}^{T} O(\frac{1}{t^2}) = O(\log T).$$

Meanwhile, since the total number of malicious participants in $M_A^1$ meets that $|M_A^1| < \frac{1}{2}M - 1$, and the consensus protocol runs $M$ participants with each participant as a commander, the consensus always succeeds with probability at least $1 - \frac{1}{l^T}$. This immediately gives us that based on (16)

$$E[T_1] = \sum_{m \in M_H} \sum_{k=1}^{K} \Delta_k E[n_{m,k}(t)] + |M_H| K l^{T-1}$$

Meanwhile, the statistical property of $n_{m,k}(t)$ depends on that of the global estimator $\tilde{\mu}_k(t)$ by our decision and update rule. The computation of $\tilde{\mu}_k(t)$ depends on set $B_t$, which is the same as the case where there are only at most $\frac{1}{2}M - 1$ malicious participants. Subsequently, we obtain

$$E[n_{m,i}(t)] \leq l + \sum_{t=L+1}^{T} P_t + \sum_{t=L+1}^{T} P_t + 0$$

$$\leq l + \frac{\pi^2}{3}$$

$$= O(\log T).$$

when $l \geq \lceil \frac{4C_1 \log T}{\Delta_i^6} \rceil$ with $\Delta_i = \mu_{i^*} - \mu_i$.

Then, based on 16, we again arrive at

$$E[T_1] \leq \sum_{m \in M_H} \sum_{k=1}^{K} \Delta_k E[n_{m,k}(t)] + |M_H| K l^{T-1}$$

$$\leq O(\log T) + |M_H| K l^{T-1}$$

Henceforth, by the regret decomposition, we have the following upper bound on the regret

$$E[R_T|A] \leq (c+1) \cdot L + E[T_1|A] + E[T_2|A]$$

$$\leq (c+1) \cdot L + O(\log T) + |M_H| K l^{T-1} \tag{20}$$

which completes the proof.

$\square$

PROOF OF THEOREM 5

*Proof.* The proof of Theorem 5 follows a similar approach to that of Theorem 4. the coordination mechanism's regret can be decomposed as follows:

$$R_T \leq L + c \cdot L + \sum_{t=L+1}^{T} \sum_{m \in M_H} (\mu_{i^*} - \mu_{a_m^t} 1_{b_t=1}) + \sum_{m \in M_H} \sum_{t=L+1}^{T} c_t 1_{h_t=1}$$

$$\doteq (c+1) \cdot L + T_1 + T_2$$

For malicious participant $j \in M_A^2$, the attacks are limited to the consensus process and do not affect the estimators. Conversely, a malicious participant $l \in M_A^1$, it targets the estimators but does not disrupt the consensus process. Given that $|M_A^1| < \frac{1}{2}M - 1$, when using Option 2, the set $B_t$ is the same as the case where only at most $\frac{1}{2}M$ participants are malicious. Therefore, we have that

$$E[T_2] \leq \log T \sum_{m \in M_H} \sum_{t=L+1}^{T} O(\frac{1}{t^2}) = O(\log T).$$

The analysis of $T_1$ requires further work, especially considering the development of this new commander selection protocol. More specifically, by definition, we have $w_m(t) = w_m = 1 - \frac{\log T}{T}$, for any $m \in M_H$. Consider the event of whether honest participant $m$ is selected as a commander as $E_m^t$. In other words, $E_m^t = 1$ if participant $m$ is a commander and 0 otherwise. Define $E_t$ as $\cap_{m \in M_H}\{E_m^t = 0\}$. Then we have that

$$E[\sum_{t=1}^{T} E_t] = \sum_{t=1}^{T} E[\cap_{m \in M_H}\{E_m^t = 0\}]$$

$$\leq \sum_{t=1}^{T} \sum_{m \in M_H} E[\{E_m^t = 0\}]$$

$$= \sum_{t=1}^{T} \sum_{m \in M_H} (1 - w_m(t)) = \log T.$$

It implies that for the total length of having no honest commanders is at most $\log T$, there is no honest commander, which indicated that the consensus fails. In the meantime, we note that if there is a honest commander in set $S_C(t)$, then the consensus is achieved with the correct $\tilde{\mu}$, i.e. $b_t = 1$, and thus we have $E[1_{b_t=0}] \leq \frac{\log T}{T}$ and $E[\sum_{t=1}^{T} 1_{b_t=0}] \leq \log T$.

Differently, by our choice, $w_j(t) = w_j = \frac{\log \frac{|M_A|}{\eta}}{T}$, for any $j \in M_A$. Then we consider the event of whether malicious participant $j$ is selected as a commander or not, namely, $F_t^j$. Likewise, $F_t^j = 1$ if participant $j$ is a commander and 0 otherwise. Define $F_t = \cap_{j \in M_A}\{\exists s \leq t, s.t. F_s^j = 1\}$. Then we obtain

$$
\begin{aligned}
P(F_t) &= P(\cap_{j \in M_A}\{\exists s \leq t, s.t. F_s^j = 1\}) \\
&\geq 1 - \sum_{j \in M_A} P(\{\forall s \leq t, s.t. F_s^j = 0\}) \\
&= 1 - \sum_{j \in M_A} (1 - w_j)^t \\
&= 1 - |M_A|(1 - w_j)^t \\
&\geq 1 - |M_A|e^{-w_j t}
\end{aligned}
$$

By the choice of $w_j = \frac{\log \frac{|M_A|}{\eta}}{T}$, we derive that $P(F_t) \geq 1 - |M_A|e^{-w_j t} = 1 - \eta$. This means that at each time step, the malicious participants have high probability of being chosen as commanders, which provides enough incentive for them to participate, and thus implies the rationality of this probability.

Subsequently, since the total number of malicious participants in $M_A^1$ meets that $|M_A^1| < \frac{1}{2}M - 1$, and the consensus protocol runs $M$ participants with at least one honest commander, the consensus always succeeds with probability at least $1 - \frac{\log T}{T}$. Based on 14, we obtain that

$$
\begin{aligned}
E[T_1] &\leq (T - L)|M_H|\mu_{i^*} - \sum_{m \in M_H}\sum_{t=L}^{T}\sum_{k=1}^{K}\mu_k E[1_{a_m^t=k}] + \sum_{m \in M_H}\sum_{t=L}^{T}\sum_{k=1}^{K}\mu_k P(b_t = 0) \\
&\leq \sum_{m \in M_H}\sum_{k=1}^{K}\Delta_k E[n_{m,k}(t)] + |M_H|KO(\log T).
\end{aligned}
$$

Again, based on the decision rule, we have the following hold for $n_{m,i}(T)$ with $l \geq \lceil\frac{4C_1 \log T}{\Delta_i^2}\rceil$,

$$
\begin{aligned}
n_{m,i}(T) &\leq l + \sum_{t=L+1}^{T} 1_{\{a_t^m=i, n_{m,i}(t)>l\}} \\
&\leq l + \sum_{t=L+1}^{T} 1_{\{\tilde{\mu}_i^m - \sqrt{\frac{C_1 \log t}{n_{m,i}(t-1)}}>\mu_i, n_{m,i}(t-1)\geq l\}} \\
&\quad + \sum_{t=L+1}^{T} 1_{\{\tilde{\mu}_{i^*}^m + \sqrt{\frac{C_1 \log t}{n_{m,i^*}(t-1)}}<\mu_{i^*}, n_{m,i}(t-1)\geq l\}} \\
&\quad + \sum_{t=L+1}^{T} 1_{\{\mu_i+2\sqrt{\frac{C_1 \log t}{n_{m,i}(t-1)}}>\mu_{i^*}, n_{m,i}(t-1)\geq l\}}.
\end{aligned}
$$

Note that taking the expectation over $n_{m,i}(t)$ gives

$$
E[n_{m,i}(t)] \le l + \sum_{t=L+1}^{T} P(\tilde{\mu}_i^m(t) - \sqrt{\frac{C_1 \log t}{n_{m,i}(t)}} > \mu_i, n_{m,i}(t-1) \ge l)
$$

$$
+ \sum_{t=L+1}^{T} P(\tilde{\mu}_i^m(t) + \sqrt{\frac{C_1 \log t}{n_{m,i}(t)}} < \mu_i, n_{m,i}(t-1) \ge l)
$$

$$
+ \sum_{t=L+1}^{T} P(\mu_i + 2\sqrt{\frac{C_1 \log t}{n_{m,i}(t-1)}} > \mu_{i^*}, n_{m,i}(t-1) \ge l) \tag{21}
$$

Using the concentration inequality, we obtain that

$$
P(\tilde{\mu}_i^m(t) + (\frac{C_1 \log t}{n_{m,i}(t)})^{\frac{1}{6}} < \mu_i, n_{m,i}(t-1) \ge l)
$$

$$
\le P(|\tilde{\mu}_i^m(t) - \mu_i| \ge O(\frac{\log t^{\frac{1}{6}}}{n_i(t)^{\frac{1}{3}}}), n_{m,i}(t-1) \ge l)
$$

$$
= P(Dist(\tilde{\mu}_i^m(t), \mu_i) \ge O(\frac{\eta_t^2}{n_i(t)}), n_{m,i}(t-1) \ge l)
$$

$$
\le P_t = \frac{1}{t^2}.
$$

Likewise, we obtain that

$$
\sum_{t=L+1}^{T} P(\mu_i + 2(\frac{C_1 \log t}{n_{m,i}(t-1)})^{\frac{1}{6}} > \mu_{i^*}, n_{m,i}(t-1) \ge l) = 0 \tag{22}
$$

if the choice of $l$ satisfies $l \ge \lceil \frac{4C_1 \log T}{\Delta_i^6} \rceil$ with $\Delta_i = \mu_{i^*} - \mu_i$, which leads to

$$
E[n_{m,i}(t)] \le l + \sum_{t=L+1}^{T} P_t + \sum_{t=L+1}^{T} P_t + 0
$$

$$
\le l + \frac{\pi^2}{3}
$$

$$
= O(\log T).
$$

Consequently, we obtain that

$$
E[T_1] \le \sum_{m \in M_H} \sum_{k=1}^{K} \Delta_k E[n_{m,k}(t)] + |M_H| K l^{T-1}
$$

$$
\le O(\log T) + |M_H| K l^{T-1}
$$

Combining all these together, we derive the following upper bound on the expected regret

$$
E[R_T|A] \le (c+1) \cdot L + E[T_1|A] + E[T_2|A]
$$

$$
\le (c+1) \cdot L + O(\log T). \tag{23}
$$

This concludes the proof of Theorem 5.

$\square$

PROOF OF THEOREM 6

*Proof.* Again, we decompose system's regret as follows:

$$R_T \leq L + c \cdot L + \sum_{t=L+1}^{T} \sum_{m \in M_H} (\mu_{i^*} - \mu_{a_m^t} 1_{b_t=1}) + \sum_{m \in M_H} \sum_{t=L+1}^{T} c_t 1_{h_t=1}$$

$$\doteq (c+1) \cdot L + T_1 + T_2$$

Differently, the definition of $c_t$ is a constant-based one, where $c_t = c1_{\exists m \in C_t \& m \in M_A^1}$ since the estimators in $C_t$ are used for computing $\tilde{\mu}_i^m(t)$. Note that here we do not count malicious participants in $M_A^2$ in, as these participant do not perform attacks on the estimators, i.e. having no negative effect on $\tilde{\mu}_i(t)$.

In the meantime, by the robust estimator property of the estimators in $B_t$, we obtain that

$$||\hat{\mu}_i(t) - \bar{z}_i(t)|| \leq c_\Delta \Delta^2$$

where with probability $1 - P_t$,

$$\Delta = \max_{m \in M_H} |\bar{\mu}_i^m(t) - \bar{z}_i(t)|$$

$$\leq \max_{m,j \in M_H} [|\bar{\mu}_i^j(t) - \mu_i| + |\bar{\mu}_i^m(t) - \mu_i|]$$

$$\leq 2\eta_t$$

This immediately implies that for $m \in M_H$

$$|\bar{\mu}_i^m(t) - \hat{\mu}_i| \leq |\bar{\mu}_i(t) - \bar{z}_i(t) + \bar{z}_i(t) - \hat{\mu}_i|$$

$$\leq 2\eta_t + 4(c_\Delta)\eta_t^2$$

$$\leq \frac{1}{2}\epsilon||q||$$

where the last inequality holds by the choice of $\epsilon$ and $||q||$ denotes the minimum value of the random variable following distribution $q_i^m$.

By assumption, we have that for $j \in M_A^1$,

$$f_i^j(t) = (1 - \epsilon)g_i^m(t) + \epsilon q_i^m(t)$$

where $f_i^j(t)$ represents the underlying distribution of the rewards of malicious agent $j \in M_A^1$. It is worth emphasizing that this assumption is consistent with (Dubey and Pentland, 2020), originated from the Huber's $\epsilon$-Contamination model (Huber and Ronchetti, 2011).

By taking the expectation over the distributions, we obtain that

$$\mu_j = (1 - \epsilon)\mu_i + \epsilon E[q]$$

This implies that for $j \in M_A^1$ with probability $1 - 2P_t$

$$|\bar{\mu}_i^j(t) - \hat{\mu}_i| \geq |\bar{\mu}_i^j(t) - \bar{\mu}_i^m(t) + \bar{\mu}_i^m(t) - \hat{\mu}_i|$$

$$\geq |\bar{\mu}_i^j(t) - \bar{\mu}_i^m(t)| - |\bar{\mu}_i^m(t) - \hat{\mu}_i|$$

$$\geq \epsilon||q|| - \frac{1}{2}\epsilon||q||$$

$$\geq \frac{1}{2}\epsilon||q||$$

This is to say that $j \in M_A^1$ does not belong to $C_t$, and thus implies that $c_t = 0$ for $t > L$ with probability $1 - 3P_t = 1 - \frac{3}{t^2}$, and $c_t = c$ with probability $\frac{3}{t^2}$.

Therefore we have that

$$E[T_2] \leq \sum_{m \in M_H} \sum_{t=L+1}^{T} O(\frac{3}{t^2}) = O(1).$$

Based on (17), we again obtain that

$$E[n_{m,i}(t)] \leq l + \sum_{t=L+1}^{T} P(\tilde{\mu}_i^m(t) - \sqrt{\frac{C_1 \log t}{n_{m,i}(t)}} > \mu_i, n_{m,i}(t-1) \geq l)$$

$$+ \sum_{t=L+1}^{T} P(\tilde{\mu}_i^m(t) + \sqrt{\frac{C_1 \log t}{n_{m,i}(t)}} < \mu_i, n_{m,i}(t-1) \geq l)$$

$$+ \sum_{t=L+1}^{T} P(\mu_i + 2\sqrt{\frac{C_1 \log t}{n_{m,i}(t-1)}} > \mu_{i^*}, n_{m,i}(t-1) \geq l) \qquad (24)$$

By the fact that with probability $1 - 3P_t$, $c_t = 0$, we again have that the validated estimator $\tilde{\mu}_i(t)$ can be expressed as with probability $1 - 3P_t$

$$\tilde{\mu}_i(t) = \sum_{j \in A_t \cap M_H} w_{j,i}(t) \bar{\mu}_i^j(t)$$

which is also equivalent to $\tilde{\mu}_i^m(t)$. Here the weight $w_{j,i}(t)$ meets the condition

$$\sum_{j \in A_t \cap M_H} w_{j,i}(t) = 1,$$

which immediately implies that

$$E[\tilde{\mu}_i(t)] = \mu_i.$$

We note that the variance of $\tilde{\mu}_i(t)$, $var(\tilde{\mu}_i(t))$, satisfies that, with probability $1 - 3P_t$

$$var(\tilde{\mu}_i(t)) = var(\sum_{j \in A_t \cap M_H} w_{j,i}(t) \bar{\mu}_i^j(t))$$

$$\leq |A_t \cap M_H| \sum_{j \in A_t \cap M_H} w_{j,i}(t)^2 var(\bar{\mu}_i^j(t)))$$

$$\leq |A_t \cap M_H| \sum_{j \in A_t \cap M_H} w_{j,i}^2(t) \sigma^2 \frac{1}{n_{j,i}(t)}$$

$$\leq |A_t \cap M_H| \sum_{j \in A_t \cap M_H} w_{j,i}^2(t) \sigma^2 \frac{k_i}{n_{m,i}(t)}$$

$$= |M_H| \frac{k_i}{n_{m,i}(t)} \sum_{j \in M_H} w_{j,i}^2(t) \sigma^2$$

$$\leq |M_H| \frac{k_i \sigma^2}{n_{m,i}(t)}$$

where the inequality holds by the Cauchy-Schwarz inequality, the second inequality holds by the definition of sub-Gaussian distributions, the third inequality results from the construction of $A_t$, and the last inequality is as a result of $(a + b)^2 \geq a^2 + b^2$.

Subsequently, we have that

$$P(\tilde{\mu}_i^m(t) - \sqrt{\frac{C_1 \log t}{n_{m,i}(t)}} > \mu_i, n_{m,i}(t-1) \geq l)$$

$$\leq \exp\{-\frac{(\sqrt{\frac{C_1 \log t}{n_{m,i}(t)}})^2}{2var(\tilde{\mu}_i^m)}\}$$

$$\leq (\exp\{-\frac{(\sqrt{\frac{C_1 \log t}{n_{m,i}(t)}})^2}{2|M_H|\frac{k_i \sigma^2}{n_{m,i}(t)}}\})(1 - 3P_t) + 3P_t$$

$$= (1 - 3P_t)\exp\{-\frac{C_1 \log t}{2|M_H|k_i \sigma^2}\} + 3P_t \leq \frac{4}{t^2} \qquad (25)$$

where the first inequality holds by Chernoff bound, the second inequality is derived by plugging in the above upper bound on $var(\tilde{\mu}_i^m(t))$, and the last inequality results from then choice of $C_1$ that satisfies $\frac{C_1}{6|M_H|k_i\sigma^2} \geq 1$.

Likewise, by symmetry, we have

$$P(\tilde{\mu}_i^m(t) + \sqrt{\frac{C_1 \log t}{n_{m,i}(t)}} < \mu_i, n_{m,i}(t-1) \geq l) \leq \frac{4}{t^2}. \tag{26}$$

This immediately implies that

$$E[n_{m,i}(t)] \leq l + \sum_{t=L+1}^{T} 4P_t + \sum_{t=L+1}^{T} 4P_t + 0$$
$$\leq l + \frac{4\pi^2}{3}$$
$$= O(\log T).$$

Then we arrive at

$$E[T_1] \leq \sum_{m \in M_H} \sum_{k=1}^{K} \Delta_k E[n_{m,k}(t)] + |M_H|Kl^{T-1}$$
$$\leq O(\log T) + |M_H|Kl^{T-1}$$

Once again, by the regret decomposition, we obtain that

$$E[R_T] \leq E[(c+1) \cdot L + T_1 + T_2]$$
$$\leq (c+1) \cdot L + O(1) + O(\log T) + |M_H|Kl^{T-1}$$
$$= O(\log T)$$

$\square$

PROOF OF THEOREM 7

*Proof.* As before, the regret is decomposed as

$$R_T \leq L + c \cdot L + \sum_{t=L+1}^{T} \sum_{m \in M_H} (\mu_{i^*} - \mu_{a_m^t} 1_{b_t=1}) + \sum_{m \in M_H} \sum_{t=L+1}^{T} c_t 1_{h_t=1}$$
$$\doteq (c+1) \cdot L + T_1 + T_2$$

We first show the monotonicity of the reputation score after the burn-in period. Recall that the reputation score of participant $i$ is defined as

$$U_i^t = \sum_{j=1}^{K} -(\bar{\mu}_j^i(t) - \tilde{\mu}_j(t))^2 - \epsilon^2 e^{(\overset{\Delta^i}{\mu}_j(t) - \tilde{\mu}_j(t)^2)}$$
$$\doteq U_i^{1,t} + U_i^{2,t}$$

where $\overset{\Delta^i}{\mu}_j(t)$ denotes the estimator for arm $j$ given by participant $i$ after the consensus step, and $\bar{\mu}_j^i(t), \tilde{\mu}_j(t)$ are the aforementioned estimators for arm $j$.

We consider $t > L$, where

$$P(\tilde{\mu}_i^m(t) + \sqrt{\frac{C_1 \log t}{n_{m,i}(t)}} < \mu_i, n_{m,i}(t-1) \geq l) \leq \frac{1}{t^2}. \tag{27}$$

We consider malicious participant $j \in M_A^1$ and honest participant $m \in M_H$, and by definition, it only attacks the estimators, which immediately gives us that

$$U_j^{2,t} = U_i^{2,t} = 0.$$

Again, by this definition and the pre-fixed $\epsilon$ zone, we obtain

$$f_i^j(t) = (1 - \epsilon)g_i^m(t) + \epsilon q_i^m(t)$$

and thus $j \in M_A^1$ with probability $1 - 2P_t$

$$\begin{aligned}
|\bar{\mu}_i^j(t) - \hat{\mu}_i| &\geq |\bar{\mu}_i^j(t) - \bar{\mu}_i^m(t) + \bar{\mu}_i^m(t) - \hat{\mu}_i| \\
&\geq |\bar{\mu}_i^j(t) - \bar{\mu}_i^m(t)| - |\bar{\mu}_i^m(t) - \hat{\mu}_i| \\
&\geq \epsilon ||q|| - \frac{1}{2}\epsilon ||q|| \\
&\geq \frac{1}{2}\epsilon ||q||
\end{aligned}$$

Subsequently, we arrive at

$$|\bar{\mu}_i^j(t) - \tilde{\mu}_i| \geq \frac{1}{2}\epsilon ||q||$$

Meanwhile, we have

$$\begin{aligned}
|\bar{\mu}_i^m(t) - \hat{\mu}_i| &\leq |\bar{\mu}_i(t) - \bar{z}_i(t) + \bar{z}_i(t) - \hat{\mu}_i| \\
&\leq 2\eta_t + 4(c_\Delta)\eta_t^2 \\
&\leq \frac{1}{2}\epsilon ||q||
\end{aligned}$$

which also implies that

$$|\bar{\mu}_i^m(t) - \tilde{\mu}_i| \geq \frac{1}{2}\epsilon ||q||$$

That is to say, the first term in the reputation score meets that

$$U_j^{1,t} \leq U_m^{1,t}$$

and subsequently, we obtain

$$U_j^t \leq U_m^t.$$

Next, let us consider malicious participant $k \in M_A^2$ and honest participant $m \in M_H$. By definition, participant $k$ only attacks the consensus process without altering the estimators. However, Equivalently, this does not imply

$$U_k^{1,t} = U_m^{1,t} = 0$$

since $\bar{\mu}_i^m \neq \bar{\mu}_i^k$ due to the randomness, which brings additional challenge.

We consider the difference between the estimators,

$$\begin{aligned}
|U_k^{1,t} &- U_m^{1,t}| \\
&\leq |\bar{\mu}_i^k(t) - \tilde{\mu}_i(t)|^2 + |\bar{\mu}_i^m(t) - \tilde{\mu}_i(t)|^2 \\
&\leq \frac{1}{2}(\epsilon ||q||)^2.
\end{aligned}$$

In the meantime, if participant $k$ serves as a validator, we immediately have

$$\begin{aligned}
\overset{\Delta^i}{\mu}_j(t) &- \tilde{\mu}_j(t)^2) > 0, \\
U_k^{2,t} &< -\epsilon^2
\end{aligned}$$

while in the meantime, $U_m^{2,t} = 0$.

Consequently, we arrive at

$$\begin{aligned}
U_k^t - U_m^t &= U_k^{1,t} - U_m^{1,t} + U_k^{2,t} - U_m^{2,t} \\
&\leq |U_k^{1,t} - U_m^{1,t}| + U_k^{2,t} \\
&\leq \frac{1}{2}\epsilon^2 - \epsilon^2 = -\frac{1}{2}\epsilon^2 < 0
\end{aligned}$$

where the second last inequality holds by assuming $||q|| \leq 1$ without loss of generality.

Combining these all together, we obtain that

$$U_j^t < U_m^t$$

for any malicious participant $j \in M_A$ and honest participant $m \in M_H$, which implies the monotonocity of $U$ quantity in the reputation score.

Subsequently, by the monotone preserving property of function $G(\cdot)$, we immediately have

$$G(U_j^t) < G(U_m^t)$$

for any malicious participant $j \in M_A$ and honest participant $m \in M_H$.

Based on the Validator selection Protocol where the top $N$ participants are selected with $|M_H| < N < 2|M_H| - 1$, we obtain that $M_H \subset S_V(t)$ and $|S_V(t)| \leq 2|M_H| - 1$, which implies that the consensus always achieves if every validator is selected as a commander for exactly once, i.e. $b_t = 1$ with probability at most $1 - Ml^{-T}$.

Otherwise, if a participant $k \in M_A^2$ is never selected as a validator, then the set of validators does not contain any malicious participants attacking the consensus, and then the consensus always achieves, i.e. $b_t = 1$.

To summarize, we have that

$$P(b_t = 1) \geq 1 - Ml^{-T}.$$

Note that the set $B_t, C_t$ herein is the same as the set of $B_t, C_t$ as in Theorem 6, which immediately implies that $j \in M_A^1$ does not belong to $C_t$, and thus implies that $c_t = 0$ for $t > L$ with probability $1 - 3P_t = 1 - \frac{3}{t^2}$, and $c_t = c$ with probability $\frac{3}{t^2}$.

Therefore we again obtain that by the definition of $T_2$ that depends on $b_t$ and $c_t$

$$E[T_2] \leq \sum_{m \in M_H} \sum_{t=L+1}^{T} O(\frac{3}{t^2}) = O(1).$$

Again, using (17), we obtain the following decomposition

$$\begin{aligned}
E[n_{m,i}(t)] \leq l &+ \sum_{t=L+1}^{T} P(\tilde{\mu}_i^m(t) - \sqrt{\frac{C_1 \log t}{n_{m,i}(t)}} > \mu_i, n_{m,i}(t-1) \geq l) \\
&+ \sum_{t=L+1}^{T} P(\tilde{\mu}_i^m(t) + \sqrt{\frac{C_1 \log t}{n_{m,i}(t)}} < \mu_i, n_{m,i}(t-1) \geq l) \\
&+ \sum_{t=L+1}^{T} P(\mu_i + 2\sqrt{\frac{C_1 \log t}{n_{m,i}(t-1)}} > \mu_{i^*}, n_{m,i}(t-1) \geq l)
\end{aligned} \qquad (28)$$

Furthermore, with probability $1 - 3P_t$, $c_t = 0$ again implies that the validated estimator $\tilde{\mu}_i(t)$ has the following explicit formula, with probability $1 - 3P_t$

$$\tilde{\mu}_i(t) = \sum_{j \in A_t \cap M_H} w_{j,i}(t)\bar{\mu}_i^j(t)$$

which is the value of $\tilde{\mu}_i^m(t)$ as well, where $w_{j,i}(t)$ are the weights such that

$$\sum_{j \in A_t \cap M_H} w_{j,i}(t) = 1.$$

This immediately gives us that

$$E[\tilde{\mu}_i(t)] = \mu_i.$$

We note that the variance of $\tilde{\mu}_i(t)$, $var(\tilde{\mu}_i(t))$, satisfies that, with probability $1 - 3P_t$

$$var(\tilde{\mu}_i(t)) = var(\sum_{j \in A_t \cap M_H} w_{j,i}(t)\bar{\mu}_i^j(t))$$

$$\leq |A_t \cap M_H| \sum_{j \in A_t \cap M_H} w_{j,i}(t)^2 var(\bar{\mu}_i^j(t)))$$

$$\leq |A_t \cap M_H| \sum_{j \in A_t \cap M_H} w_{j,i}^2(t)\sigma^2 \frac{1}{n_{j,i}(t)}$$

$$\leq |A_t \cap M_H| \sum_{j \in A_t \cap M_H} w_{j,i}^2(t)\sigma^2 \frac{k_i}{n_{m,i}(t)}$$

$$= |M_H| \frac{k_i}{n_{m,i}(t)} \sum_{j \in M_H} w_{j,i}^2(t)\sigma^2$$

$$\leq |M_H| \frac{k_i\sigma^2}{n_{m,i}(t)}$$

where the inequality holds by the Cauchy-Schwarz inequality, the second inequality holds by the definition of sub-Gaussian distributions, the third inequality results from the construction of $A_t$, and the last inequality is as a result of $(a + b)^2 \geq a^2 + b^2$.

Subsequently, we have that

$$P(\tilde{\mu}_i^m(t) - \sqrt{\frac{C_1 \log t}{n_{m,i}(t)}} > \mu_i, n_{m,i}(t-1) \geq l)$$

$$\leq \exp\{-\frac{(\sqrt{\frac{C_1 \log t}{n_{m,i}(t)}})^2}{2var(\tilde{\mu}_i^m)}\}$$

$$\leq (\exp\{-\frac{(\sqrt{\frac{C_1 \log t}{n_{m,i}(t)}})^2}{2|M_H|\frac{k_i\sigma^2}{n_{m,i}(t)}}\})(1 - 3P_t) + 3P_t$$

$$= (1 - 3P_t)\exp\{-\frac{C_1 \log t}{2|M_H|k_i\sigma^2}\} + 3P_t \leq \frac{4}{t^2} \tag{29}$$

where the first inequality holds by Chernoff bound, the second inequality is derived by plugging in the above upper bound on $var(\tilde{\mu}_i^m(t))$, and the last inequality results from then choice of $C_1$ that satisfies $\frac{C_1}{2|M_H|k_i\sigma^2} \geq 1$.

In a similar manner, we obtain

$$P(\tilde{\mu}_i^m(t) + \sqrt{\frac{C_1 \log t}{n_{m,i}(t)}} < \mu_i, n_{m,i}(t-1) \geq l) \leq \frac{4}{t^2}. \tag{30}$$

Plugging the concentration-type inequalities in, we derive

$$E[n_{m,i}(t)] \leq l + \sum_{t=L+1}^{T} 4P_t + \sum_{t=L+1}^{T} 4P_t + 0$$

$$\leq l + \frac{4\pi^2}{3} = O(\log T).$$

Then we arrive at

$$E[T_1] \le \sum_{m \in M_H} \sum_{k=1}^{K} \Delta_k E[n_{m,k}(t)] + |M_H| K l^{T-1}$$
$$\le O(\log T) + |M_H| K l^{T-1}$$

Once again, by the regret decomposition, we obtain that

$$E[R_T] \le E[(c+1) \cdot L + T_1 + T_2]$$
$$\le (c+1) \cdot L + O(1) + O(\log T) + |M_H| K l^{T-1}$$
$$= O(\log T)$$

$\square$

## F    DISCUSSIONS

**Block Information**

Let $h_j^t(\mathcal{F}_t)$ be the estimators given by malicious participant $j \in M_A$ where $\mathcal{F}_t$ denotes the history up to time step $t$ (everything on the blockchain and additional information shared by other participants). The blocks on the blockchain record the execution information. Specifically, at each time step $t$ the block records the global estimators $\{\tilde{\mu}_i(t)\}_i$ and local estimators $\{\bar{\mu}_i^m(t)\}_{m,i}$, counts $\{n_{m,i}(t)\}_{m,i}$, $B_t$ specified in Aggregation, and arms $a_m^t$ pulled. Moreover, the block also records the reward $r_i^m(t)$ of each participant $m \in M$. The information related to an individual participant, such as $\bar{\mu}_i^m(t)$ and $r_i^m(t)$, is signed by the participants using digital signatures that are the same across time. Each quantity in the block related to $m$ is stored together with the public key of client $m$. Arm indices also need to be stored for quantities depending on $i$. If it is desirable the arms to be pseudo anonymous, public keys of arms can be used and digital signatures would be created based on private key pairs of (participant, arm). By using Global Update in Algorithm 4, the definition of $\{\tilde{\mu}_i(t)\}_i$ and (1), all these quantities can be verified given $r_i^m$ and $a_m^t$.

**Ratinality of $R_T$**

We argue the rationale of this regret definition as follows. It holds true that these two regret measures are well-defined, considering that $M_H$ is fixed and does not change with time. Furthermore, our definition aligns with those used in the context of blockchain-based federated learning (Zhao et al., 2020), as their objective is to optimize the model maintained by honest participants, though without involving online decision making. Additionally, this definition is consistent with the existing robust multi-agent MAB problem (Vial et al., 2021), except that the cost mechanism is introduced which incentives participation and guarantees correctness. Compared to the multi-agent MAB, our regret is averaged over only honest participants due to the existence of malicious participants. Note that the two measures are the same if the number of malicious participants is zero since the cost $c_t$ is also zero in such a case, implying consistency.

**Discussion on Theorem 1**

It is worth noting that the minimum number of honest participants is consistent with (Zhu et al., 2023). Although they establish the regret bound in a cooperative bandit setting with Byzantine attacks for any number of participants, the regret is only smaller than the individual regret when this assumption holds for every neighbor set of every honest participant at each time step. Otherwise, the regret is even larger, providing no advantage or motivation for participants to collaborate, essentially reducing the problem to the single-agent MAB problem.

**Discussion on Theorem 2**

We would like to emphasize that there should be at least one honest commander who has the same sent estimator as the honest validators. The honest validators choose to do majority voting only when the received message matches their own. In other words, consensus alone is not sufficient for the protocol; rather, consensus on the correct estimators guarantees the desired functionality of the protocol.

**Discussion on Theorem 7**

**Reputation-based Validator Selection**

It is worth mentioning that the reputation system also ensures the fairness of the protocol, or equivalently, decentralization, as no single participant is favored and the criterion is merit-based, depending on how much they contribute to the protocol. Also, privacy is maintained since the participants are not aware of $U_i^t$ due to the existence of $G(\cdot)$. Meanwhile, the number of validators $N$ given by the reputation score system is flexible in the range of $[M_H, 2M_H - 1]$, balancing the trade-off between decentralization and efficiency. While it is practically meaningful, it is also crucial to demonstrate the theoretical effectiveness of the coordination mechanism after incorporating the reputation score system. Subsequently, we present the following theoretical regret guarantee of the entire system with the above reputation score system for validator selection. The formal statement reads as follows.

It is worth noting that existing works, such as (Dennis and Owen, 2015; Zhou et al., 2021; Arshad et al., 2022), have proposed reputation-based validator selection. However, most of this work focuses on the practical performance of a reputation system, with limited theoretical analyses on the security guarantee. Here, we prove that the reputation system ensures optimal regret, which is only obtainable when the coordination mechanism is secure enough in terms of the consensus and the associated information after the consensus.

$\square$

