# OpenReview forum: "Decentralized Blockchain-based Robust Multi-agent Multi-armed Bandit"
_ICLR.cc/2025/Conference — ICLR 2025 Conference Withdrawn Submission_

### Official Review · Reviewer_X59D · 2024-10-19

**Soundness:** 3
**Presentation:** 2
**Contribution:** 4
**Rating:** 5
**Confidence:** 3

**Summary:**

This paper introduces the blockchain technique to multi-agent multi-armed bandits to improve the algorithm's robustness. The authors propose a new algorithmic framework called BC-UCB. Afterwards, they analyze the algorithm under different conditions, such as the impact of different numbers of malicious agents and different costs on the results.

**Strengths:**

1. The idea of ​​this paper is very novel: it is interesting to apply blockchain to the MAB problem. As the authors noted, this technique can simultaneously address issues of optimality, security, and privacy.
2. The analysis is concrete. The careful consideration of various scenarios makes the theoretical results of this paper highly detailed.

**Weaknesses:**

I have several concerns:
1. The first main concern is the writing of this paper. The introduction section is excessively long, but still does not introduce the insight of the problem clearly. In many places, the author lacks examples or explanations. For example, in section 3, the author states a lengthy method, but does not provide intuition or insight, making it easy for people to get lost in the details and not understand what the sub-technique is used for.
2. No numerical results. This paper has no numerical results at all, making it difficult to intuitively assess the effectiveness of the BC-UCB algorithm. In fact, there are a series of works considering the robustness of the algorithms against Byzantine attacks. The authors can consider using them as baselines for experiments.

**Questions:**

Please try to solve the problems in the weakness part. In addition, I have some extra questions:
* Is there any result about the lower bound of the regret? The difficulty of the problem is unclear, thus the efficiency of the algorithm cannot be evaluated reasonably.
* Can the authors compare their results with other Byzantine-robust algorithms, both theoretically and numerically? I am concerned that the introduction of blockchain (although the authors claim that the problems like privacy can be solved meanwhile) may increase the complexity of the problem. If the improvement in regret is not significant enough, then this technology may be difficult to apply.

---

### Official Review · Reviewer_bJu9 · 2024-11-02

**Soundness:** 2
**Presentation:** 1
**Contribution:** 2
**Rating:** 3
**Confidence:** 3

**Summary:**

To the extent that I was able to understand, the paper studies a problem of multi-agent, multi-arm bandits (MABs) on blockchains. There are honest and malicious participants and a centralised mechanism that aims to maximise the honest participants' rewards. The honest participants are trying to reach consensus, if I understood correctly, through an algorithm that is shown to have (logT) bounded regret which is consistent with MAB settings without malicious agents.

**Strengths:**

- The paper showcases significant effort.
- Its claims are backed by theoretical proofs.

**Weaknesses:**

- Regarding its readability, the paper needs to be significantly streamlined. For instance, the paper spends 2.5 pages on various topics and only lines 151-158 to describe its contributions. I found it very hard to read and understand. Also, I think that paper ends abruptly at the page limit and then simply continues in the appendix (the conclusions are in page 15). Other examples include the caption of Figure 1 which almost overlaps with the text and the storm of references (especially in the first part) which don't really provide a context for the paper but rather confuse the reader.
- Regarding its rigor, there seem to be undefined abbreviations (e.g., UCB), redundant notations, and questionable assumptions, e.g., in line 169 why is B=T and how can we assume that the number of blocks is fixed (and not growing) and each block is valid with some probability? This is an uncommon representation of a blockchain (history). Other examples include Theorem 1 which mentions "existential forgery, and universal composability" but which are only defined much later in the appendix (same for the other theorems). This significantly interrupts the flow.
- The paper mingles several notions, federated learning, blockchains, multi-arm bandits, multi-agent learning etc and to allow for a reasonable assessment, it needs to make a stronger effort to both clarify this combination and to draw clear connections to existing literature and existing real-world (or at least theoretical) problems.
- After reading the problem formulation a couple of times, I still struggle to understand what is the context: what are multi-agent MABs on blockchains? Is there a practical scenario that this setting captures?

**Questions:**

Can the authors address the weaknesses mentioned above? In particular, can the others describe in concise terms:
- What is the real-world or theoretical setting that they are trying to solve with their model? Can the authors provide simulations on synthetic or real-data to demonstrate this?
- What is/are the closest related paper/s? In particular, at the end of the abstract, there seem to be references to existing work with logT regret in similar (but less adversarial) settings. Can the authors elaborate on these comparisons (with concrete references)?
- Why is the purpose of the proposed algorithm? What is its computational complexity and is it centralised or decentralised?
- What do the arms stand for in this scenario and what is the utility that the centralised mechanism is trying to optimise for the honest participants?
- Is this applicable to any blockchain or specifically the Ethereum blockchain (which runs with validators)?

---

### Official Review · Reviewer_5Xpj · 2024-11-04

**Soundness:** 3
**Presentation:** 2
**Contribution:** 2
**Rating:** 3
**Confidence:** 2

**Summary:**

The paper incorporates techniques from blockchains to design a cooperative decision making framework for multi-agent MAB problems with malicious participants. The proposed framework has been shown to satisfy multiple criteria, including optimality (with proven regret bounds), security, and privacy.

**Strengths:**

+ The paper leverages blockchains in cybersecurity to complement existing robust multi-agent MAB literature, addressing the three overlooked concerns: optimality, security, and privacy during cooperative decision making.
+ Detailed analyses of theoretical guarantees with proofs for different problem settings e.g., the number of malicious participants, the cost definition, etc.

**Weaknesses:**

- The motivations and literature review for robust multi-agent MAB problem and blockchains as well as their incorporation are not sufficient and hard to follow. There should be a section dedicated to related work for these respective topics, instead of a 3-page introduction with no headings.
- Description and motivations on the considered types of attacks (e.g., Byzantine) from participants are not clear. This information is particularly important given that the proposed framework relies on the assumptions about the structure of malicious behaviors as stated by the authors.
- As a result of the previous two points, the practicality and implications of the proposed framework are not clear.
- Missing citations throughout the paper:
	Lines 164-165: standard notations for traditional MAB and multi-agent MAB
	Line 224: "... existing work on Byzantine-resilient multi-agent MAB"
	Line 230: "This assumption is quite common in blockchain works."

**Questions:**

Beside addressing the above weaknesses, please clarify the considered malicious behaviors from participants with justifications.

---

### Official Review · Reviewer_3uQ2 · 2024-11-20

**Soundness:** 3
**Presentation:** 2
**Contribution:** 2
**Rating:** 3
**Confidence:** 4

**Summary:**

The paper introduces a framework that combines Multi-armed bandits with blockchains and smart contracts to address multi-agent settings where some number of agents could be malicious. The rewards are only obtainable when the coordination mechanism's security is guaranteed which is controlled by some smart contract. The authors provide provable regret guarantees for their framework and argue about other advantages such as security and privacy.

**Strengths:**

- The paper combines ideas from MAB literature with tools from blockchains/smart contracts which is an interesting innovation.
- The authors provide detailed theoretical analyses of the regret.

**Weaknesses:**

- The authors could better motivate the problem. The described applications/examples do not feel like a good fit. E.g. in many cases the other agents are merely self-interested.
- The paper does not include any experimental/numerical results.
- The paper does not seem to offer any insights that could be of independent technical interest either from MAB or blockchain side but feels more like a collage of largely pre-existing techniques. Although this is not a deal breaker, it makes the lack of sufficient motivation even more pressing as a problem.

**Questions:**

- Is there a clear real world motivated problem that this setting is making progress on? A single well described setting is much more effective than a sprawling light weight description of many settings.
- In such a pre-existing setting, what are the current solutions and it is possible to argue theoretically/experimentally that this frameworks significantly improves upon them?

---

### Note · Authors · 2024-11-27

I have read and agree with the venue's withdrawal policy on behalf of myself and my co-authors.